# Disentangling the Factors of Convergence between Brains and DINOv3

**Joséphine Raugel**[1,2]     **Marc Szafraniec**[1]     **Huy V. Vo**[1]     **Camille Couprie**[1]

**Patrick Labatut**[1]     **Piotr Bojanowski**[1]     **Valentin Wyart**[2]     **Jean-Rémi King**[1]

[1]**Meta AI**     [2]**Ecole Normale Supérieure, PSL Research University**
`{jeanremi, josephiner}@meta.com`

## Abstract

Many AI models trained on natural images develop representations that resemble those of the human brain. However, the factors driving this brain-model similarity remain poorly understood. To disentangle how the model, training and data independently lead a neural network to develop brain-like representations, we train a family of self-supervised vision transformers (DINOv3) that systematically vary these factors. We compare their representations of images to those of the human brain recorded through fMRI and MEG, providing high resolution in both spatial and temporal analyses. We assess the brain-model similarity with three complementary metrics focusing on representational similarity, topographical organization, and temporal dynamics. We show that all three factors - model size, training amount, and image type - independently and interactively impact each of these brain similarity metrics. In particular, the largest DINOv3 models trained with the most human-centric images reach the highest brain-similarity. These findings generalize across seven additional models. This emergence of brain-like representations in AI models follows a specific chronology during training: models first align with the early representations of the sensory cortices, and only align with the late and prefrontal representations of the brain with considerably more training. Finally, this developmental trajectory is indexed by structural and functional properties of the human cortex: representations acquired last by the models specifically align with cortical areas with the largest developmental expansion, thickness, least myelination and slowest timescales. Overall, these findings disentangle the interplay between architecture and experience in shaping how artificial neural networks come to see the world as humans do, thus offering a promising framework to understand how the human brain comes to represent its visual world.

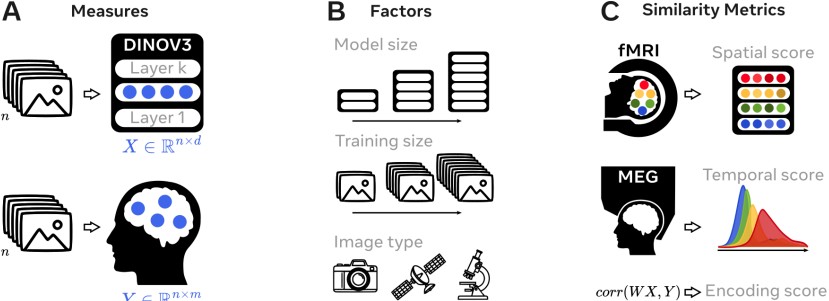

**Method Figure. A.** We compare the activations of DINOv3 to the activations of the human brain in response to the same images. **B.** To understand the factors that steer DINOv3 towards brain activity, we train *from scratch* a variety of models on different image domains (human-centric, satellite or biological data), with varying amounts of data. **C.** We compare each model to both fMRI and MEG (high spatial and temporal resolutions) by computing the linear similarity of their representations and the similarity of their hierarchical organization (encoding, spatial and temporal scores).

# 1 INTRODUCTION

**Brain-AI similarity.** Deep learning has transformed computer vision over the past decade. State-of-the-art deepnets now achieve human-level or superior performance across a variety of tasks including classification (Siméoni et al., 2025; Tschannen et al., 2025), object detection (Redmon et al., 2016), semantic segmentation (Cheng et al., 2022), and medical image analysis (Esteva et al., 2017; Lorenci et al., 2025). Surprisingly, the internal representations of these deep learning models appear to be related to those of the human brain: multiple electrophysiology (Yamins et al., 2014a; Yamins & DiCarlo, 2016; Schrimpf et al., 2018; Zhuang et al., 2021), functional Magnetic Resonance Imaging (Eickenberg et al., 2017; Millet et al., 2023; Doerig et al., 2025; Tang et al., 2023; Nikolaus et al., 2024), magneto-encephalography studies (Cichy et al., 2016; Seeliger et al., 2018; Caucheteux & King, 2022; Banville et al., 2025) have now consistently shown that the activation patterns of these models linearly map onto those of the cortex in response to the same images.

**Theoretical importance.** Understanding the principles at the origin of this representational similarity between AI models and the human brain is of primary importance, to understand the laws of information processing that may be universally shared across neural networks. Indeed, several lines of research (Hasson et al., 2020; Huh et al., 2024; van Rossem & Saxe, 2024; Cagnetta et al., 2024; Mehrer et al., 2020; Mahner et al., 2025; Simkova et al., 2025) suggest that there exists universal principles that constrain the structure and emergence of representations in neural networks.

**Challenge: Unclear causes.** The precise factors responsible for the representational similarity between computer vision models and the human remain currently unclear. This gap of knowledge is partly due to the fact that previous studies primarily focused on pretrained networks that *simultaneously* vary in training objectives, architectures and data regime (Conwell et al., 2021; Rajesh et al., 2024). How each of these factors independently and interactively leads a model to converge to brain-like representations thus remains unclear.

To address this issue, we systematically train a variety of DINOv3 models (Siméoni et al., 2025), while independently varying their size, data type and training quantity. DINOv3 has the advantage of being self-supervised, and can thus be trained on different types of naturalistic but non-human centric and non-labelled data such as satellite images (Siméoni et al., 2025) and biological images (Lorenci et al., 2025).

Here, we compare a variety of DINOv3 models to the brain responses to images, as recorded with ultra high field (7T) functional MRI and magneto-encephalography (MEG) to get a high spatial and temporal resolution of the cortical representations, respectively. For this, we implement three similarity metrics. First, we use a standard linear mapping metric, often referred to as *encoding score* (Naselaris et al., 2011b), which evaluates the linear correspondence between the representations of two systems. Second, we evaluate, with fMRI, whether this linear mapping follows a similar *spatial* organization, whereby the first and last layers of the model would best match the sensory visual and prefrontal cortices, respectively. Finally we evaluate, with MEG, whether this mapping follows a similar *temporal* organization, whereby the first and last layers of the model best match the early and late MEG responses, respectively.

# 2 METHOD

## 2.1 APPROACH

We aim to identify the factors that make modern computer vision models process and represent natural images similarly to the human brain. Following previous work (Kriegeskorte et al., 2008; DiCarlo et al., 2012; King & Dehaene, 2014), we rely on the definition of "representation" as "linearly readable information". We employ the encoding analysis procedure introduced by Naselaris et al. (2011b) to evaluate the representational similarity between an AI model and brain recordings. This linear model seeks to find whether there exists a linear mapping $W \in \mathbb{R}^{m \times d}$ that reliably predicts $m$-dimensional brain activity ($Y \in \mathbb{R}^{n \times m}$) given the $d$-dimensional model activation ($X \in \mathbb{R}^{n \times d}$) in response to $n$ images:

$$\arg\min_W \left\{ \|Y - XW\|_2^2 + \lambda\|W\|_2^2 \right\}$$

with $\lambda$ the ridge regularization parameter. We use linear probes to maintain geometries of compared representations. We use scikit-learn's `RidgeCV` (Pedregosa et al., 2011), 10 logarithmically-spaced regularization $\lambda$ in between $10^0$ and $10^8$, and a 5-split cross-validation.

## 2.2 METRICS

**Encoding score** Given two representations $X$ and $Y$, we quantify their overall representational similarity by computing, for each split separately and then averaged, an *encoding score* with a Pearson correlation score $R \in [-1, 1]$:

$$R = corr(WX_{test}, y_{test})$$

We rely on encoding as the basis for our three metrics (encoding, spatial, and temporal scores), rather than decoding, as decoding metrics cannot be meaningfully compared across models with different architectures and representational spaces. Following the rationale of (Naselaris et al., 2011a), encoding provides an interpretable mapping from model features to neural responses, comparable across architectures and training regimes. For clarity, we can either summarize the average R score across brain dimensions, or plot them all separately to get information about where brain activations are linearly predictable from the model. In some analysis, we use $\tilde{R} = R/\max(R)$, the normalized encoding score, which peaks at 1.

**Spatial score** To assess whether a model organized its processing hierarchy similarly to that of a brain with a *spatial score*, we proceed in four steps. First, we evaluate an encoding score for each dimension $m$ of the brain, and from 22 layers $k \in [0, 1]$ of the model, where 0 is the first layer, and 1 is the last layer. Second, we identify the layer that best predicts this brain response: $k^*$. Third, we approximate the hierarchical position $m^*$ of each brain region, as its Euclidean distance from V1 in the standardized MNI space, in mm. Note that this is a coarse approximation, as the actual cortical hierarchy does not strictly follow such distances, and may be considerably more complex (Felleman & Van Essen, 1991). Finally, we compute the spatial score as the correlation between $m^*$ and $k^*$. For clarity we restrict these analyses to regions of interest.

**Temporal score** To evaluate an analogous metrics from MEG recordings, we estimate a *temporal score*: the correlation between the model layers $k$ and $T_{\max}^{layer}$ – the time at which each layer of the model is maximally predictive of brain activity. To limit noisy estimate, we average on the temporal window during which $\tilde{R}^k \geq 95\%$ where $\tilde{R}^k$ is the normalized encoding-score of the layer $k$.

## 2.3 MODELS

**Architecture.** DINOv3 is an open-source, state-of-the-art, self-supervised learning vision transformer model trained on 1.7 billion natural images (Siméoni et al., 2025). We train, from scratch, a selection of eight variants of this DINOv3 model to ensure a comprehensive evaluation ranging through architectures, training scale and data types.
First, we leverage the DINOv3-7B, trained across $1e^7$ checkpoints. We analyze comparatively DINO Small, Base, Large and Giant, after training for $5e^6$ training steps on 1.7B images with the same configuration. Additionally, we train and analyze comparatively 3 versions of the DINO Large architecture: DINO human, DINO Cellular and DINO Satellite. These models were configured similarly and trained, from scratch, over $5e^6$ steps on 10M images; they only differ in the type of images with which they were trained.

Table 1: Specifications of DINOv3 model variants.

| Model | Parameters | Layers | Batch | Images |
|---|---|---|---|---|
| DINOv3 | 7B | 40 | 4096 | Human centered 1.7B |
| DINOv3 Giant | 1.1B | 32 | 4096 | Human centered 1.7B |
| DINOv3 Large | 300M | 24 | 4096 | Human centered 1.7B |
| DINOv3 Base | 86M | 12 | 4096 | Human centered 1.7B |
| DINOv3 Small | 21M | 12 | 4096 | Human centered 1.7B |
| DINOv3 Human | 300M | 24 | 2048 | Human centered 10M |
| DINOv3 Cellular | 300M | 24 | 2048 | Cellular 10M |
| DINOv3 Satellite | 300M | 24 | 2048 | Satellite 10M |

## 2.4 DATASETS.

**Images.** DINOv3-7B and DINO human were trained on the same human-centric data. This dataset was constructed from a large pool of web images, street views and ImageNet (Deng et al., 2009). These images went through platform-level content moderation to prevent harmful contents, in order to obtain a data pool of approximately 17 billion images. This data pool was curated following the procedure of (Siméoni et al., 2025) to obtain a large-scale pre-training dataset of 1.7 billion images. To compare models trained with different types of images, we re-trained three distinct large DINOv3 with one of three types of natural images – human-centric, cellular and satellite images – matched in terms of quantity (10M images each).
Human-centric images correspond to the dataset used for training the original DINOv3 model. For our comparative analyses on human-centric, cellular and satellite images, we randomly selected from this dataset of 1.7 billion images a subset of 10 million images.
Cellular images correspond to the ExtendedCHAMMI dataset, which consists of fluorescent microscopic images of cells revealing cellular structures into different channels (e.g. nucleus, mitochondria, microtubules, etc.) (Lorenci et al., 2025).
Satellite images correspond to a random subset of the SAT-493M dataset, which consists of approximately 500 million images sampled randomly from Maxar RGB ortho-rectified imagery at 0.6 meter resolution (Siméoni et al., 2025).

**Magnetoencephalography (MEG).** We use the THINGS-MEG dataset (Hebart et al., 2023a), which consists of MEG recordings from four healthy participants viewing 22,500 naturalistic images, representing a total of 1,800 object concepts (Hebart et al., 2023b). Images were presented during 1.5 s, while participants maintained fixation. To limit the impact of noise we apply a band-pass filter between 0.1 and 20 Hz, down-sample the signal at 30 Hz, time-lock the brain responses to individual words, and epoch the corresponding neural data between -0.5 s and +3 s relative to word onset using MNE-Python (Gramfort et al., 2013). Finally, we z-score MEG signals across words, for each MEG channel and each time point independently.
*time ROIs*. We study individually three 0.05s-long time ROIs across the processing time of an image, to study the relative impact of each layer in the encoding of the cognitive process at play during that time. These time windows span .08-.13s, .13-.18s and .5-.55s.
$T_{max}^{layer}$. To study the dynamics of each layer, we compute $T_{\max}^{layer}$, the mean of the temporal window during which $\tilde{R}^{layer} \geq 95\%$ where $\tilde{R}^{layer}$ is the normalized encoding-score of each layer.

**Functional Magnetic Resonance Imaging (fMRI).** We leverage the Natural Scenes Dataset (Allen et al., 2022), a 7 tesla fMRI dataset which consists of recordings from eight subjects, each observing a total of 10 000 natural scenes during 4 seconds each, while performing a continuous recognition task. We encode the BOLD signal on the fsaverage surface at 5.5 s after image onset. This timestep corresponds to the peak of decoding of the image from the BOLD signal.
*Regions of interest (ROIs)*. For clarity, we select a representative set of 15 regions of interest (ROIs) spanning the anatomy of the cortex, among the regions encoded with an averaged FDR-corrected t-test $p < 0.01$, among voxels forming the ROI. These ROIs are distributed from posterior-occipital lobe to prefrontal cortex.

To investigate the cortical properties that index representational similarity, we analyze our results in light of four cortical maps, made available through Neuromaps (Markello et al., 2022):
*Cortical expansion* (Hill et al., 2010) reflects the difference of cortical surface area between infants and adults. *Myelin concentration* is estimated from the T1w/T2w ratio in the HCP S1200 dataset (Van Essen et al., 2013). *Intrinsic timescales* are derived from mapping electromagnetic networks to hemodynamic network and indexing the temporal integration window of each region (Shafiei et al., 2021). *Cortical thickness* is estimated by measuring the distance between "white" and "pial" Freesurfer surfaces (Fischl, 2012) from structural MRI in the Human Connectome Project (Van Essen et al., 2013).

## 2.5 STATISTICS

*fMRI voxels.* We only plot and analyze voxels thresholded with $p < 0.01$ after a FDR-corrected t-test. *Across subjects.* To evaluate statistical estimates across subjects, we perform a Wilcoxon test using scipy (Virtanen et al., 2020). To correct for multiple comparison, we apply a false discovery rate correction, as implemented in MNE-Python (Gramfort et al., 2013).

*Half times.* To analyze the speed of convergence of DINO models during training, we estimate the 'half time': the training step at which the similarity metric reaches half of its final value.

# 3 RESULTS

## 3.1 DINOv3-BRAIN SIMILARITY

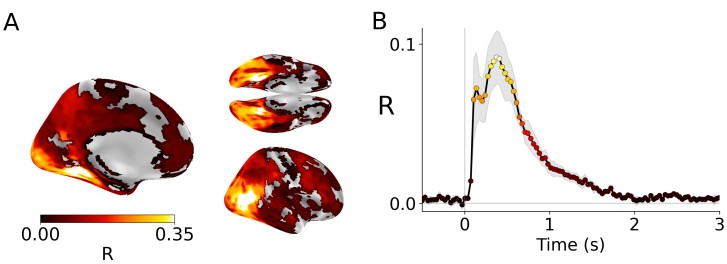

Figure 1: **Brain-DINOv3 similarity across space and time. A. Across cortical space.** Similarity between DINOv3 embedding and the fMRI responses to corresponding images as estimated with a Pearson Brain-Score, and FDR-corrected-thresholded at p < 0.01 (left: medial view of left hemisphere, top right: bottom view; bottom right: lateral view of right hemisphere). **B. Across time.** Similarity between DINOv3 embedding and MEG responses to the corresponding images. The error bar indicates the standard error of the mean across 4 subjects.

**Encoding score.** To verify that DINOv3 generates representations of natural images that are similar to those of the brain, we perform a cross-validated encoding analysis by evaluating the linear mapping between the activations of DINOv3 and of the brain in response to the same images. Functional MRI results show that DINOv3 has representations that primarily peak in the visual pathway (R=.45 $\pm$ .039 - SEM across subjects), mostly in the lateral-occipitotemporal (MT: R=.34 $\pm$ .026) and ventromedial visual cortex (VMV2: R=.28 $\pm$ .025), Fig 1A.

MEG results show that this similarity rises around 70 ms after image onset (R=.09 $\pm$ .017, Fig 1B) and remains significantly above chance level up to 3 seconds after image onset (p < 1e-3).

These results are consistent with past studies (Eickenberg et al., 2017; Schrimpf et al., 2018; Tang et al., 2025) and additionally show that areas typically discarded from the visual pathways, e.g. prefrontal regions BA 44, BA 45, IFSa and IFSp, also present activations that are linearly predictable from the AI embedding.

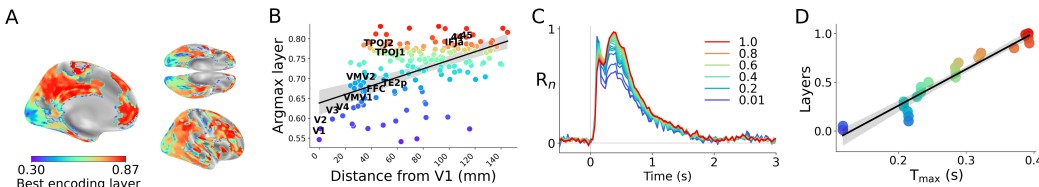

Figure 2: **The representational hierarchy of DINOv3 corresponds to the brain's. A.** Voxel-wise best encoding layers of DINOv3, FDR-corrected and thresholded at p < 0.01 (left: medial view of left hemisphere, top right: bottom view; bottom right: lateral view of right hemisphere). **B.** Plotting the correlation between the best encoding layer for each region and the euclidean distance of this region from V1, in mm. The Pearson correlation is r = 0.38, p < 1e-6. Plotted regions are encoded with FDR-corrected thresholding at p < 0.01. **C.** Dynamic brain-score across time between each layer of DINOv3 and MEG responses to the corresponding still images. **D.** Plotting the correlation between the layers and their $T_{\max}^{layer}$, in s. The Pearson correlation is r = 0.84, p < 1e-5. Plotted regions are encoded with FDR-corrected thresholding at p < 0.01.

**Spatial score.** Does the hierarchy of representations of DINOv3 correspond to the visual hierarchy in the human brain? To address this question, we estimate the "spatial score". The fMRI results

confirm that the lowest layers of DINOv3 tend to best predict the lower-level sensory regions such as V1, whereas the highest layers tend to best predict higher-level regions of the brain, such as the prefrontal cortex (Fig 2A, B). The Pearson correlation between (i) the Euclidean distance between each brain region and V1, and (ii) the best encoding layer is highly significant, R=0.38, $p < 1e^{-6}$ (Fig 2B).

**Temporal score.** To complement this fMRI "spatial score", we evaluate an MEG "temporal score". We identify the layer which best predicts each time ROIs relative to image onset in the MEG. The results show a significant correlation between layers and their $T_{\max}^{layer}$, hereafter referred to as the temporal score (Fig 2C, D). The temporal score R=0.96, $p < 1e^{-12}$, shows that the early and late layers of DINOv3 consistently align with the earliest and latest MEG responses, respectively.

**Generalization to multiple architectures.** To test for generalization of Encoding, Spatial and Temporal scores we reproduce these results on a variety of seven vision models including CNNs, supervised and self-supervised ViTs as well as vision-language contrastive transformers. We find similar scores for all three metrics, across all these models (Figs. S1,S2,S3,S4,S5)

## 3.2 What factors lead DINOv3 to become brain-like?

**Impact of training.** To clarify the emergence of brain-like processes in DINOv3, we evaluate the encoding score, spatial score and temporal score at each selected training step of DINOv3, and summarize their developmental speed with a "half time": i.e. the training step where half of the final score is reached. First, before training the encoding score reaches R=$.03 \pm 2e^{-4}$, after training it ultimately converges to R=$.09 \pm 5e^{-4}$ (Fig 3). These R-scores are averaged across voxels – the best voxel peaking at R=$.45 \pm .038$. The half time of the encoding score occurs around 2% of the training, around $2e^5$ training steps (i.e. 800 million images). Second, the temporal score emerges faster than the encoding-score: with a half time around 0.7% of the training, and a convergence at R=0.96 ($p < 1e^{-12}$). Finally, the spatial score reaches its half time later, at 4% of the training, and converges to R=0.4 ($p < 1e^{-6}$). We reproduce the spatial score using the value of each ROI along the sensory-to-transmodal gradient map from (Margulies et al., 2016) instead of the euclidean distance of this ROI from V1, see Supplementary Figure S6. We obtain a similar increase across training from -0.32 to 0.13 (1st to last checkpoint).

**Generalization to multiple architectures.** Across seven vision models with diverse architectures – CNNs, supervised ViTs, vision-language contrastive transformers, etc. - trained models consistently show higher encoding, spatial, and temporal scores than untrained models. Additionally, trained models tend to show convergent encoding, spatial, and temporal scores, whereas scores from untrained models vary more widely – likely reflecting differences in their inductive biases. These additional analyses are reported in S1,S2,S3,S4,S5, comparing trained and untrained versions of all models.

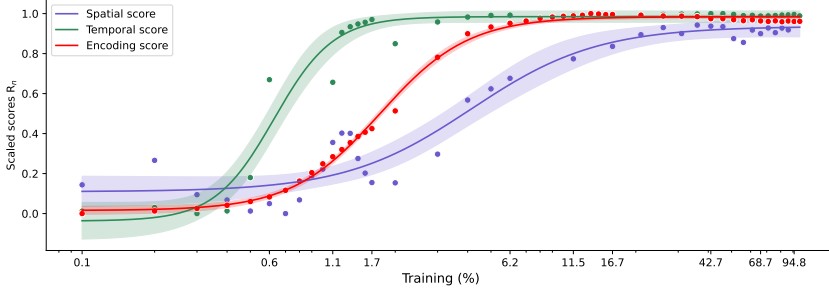

Figure 3: **Evolution of scaled temporal, encoding and spatial scores** as a function of DINOv3's training. The Evolution of these unscaled metrics is presented in Supplementary Fig. S7.

Are these developmental trajectories identical across temporal and brain regions of interest? To address this issue, we evaluate the same analyses on specific regions or temporal windows of interest. Functional MRI results show that low-level visual regions (e.g. V1, V2) are marked by lower half times of the last layer than high-level prefrontal cortices (e.g. IFSp, IFSa), Fig 4A,C,E. The correla-

tion between half time and anatomical location (coarsely defined as the Euclidean distance to V1) is R=0.91, p< $1e^{-5}$. . Similarly for MEG, earlier windows (e.g. <200 ms) are marked by lower half

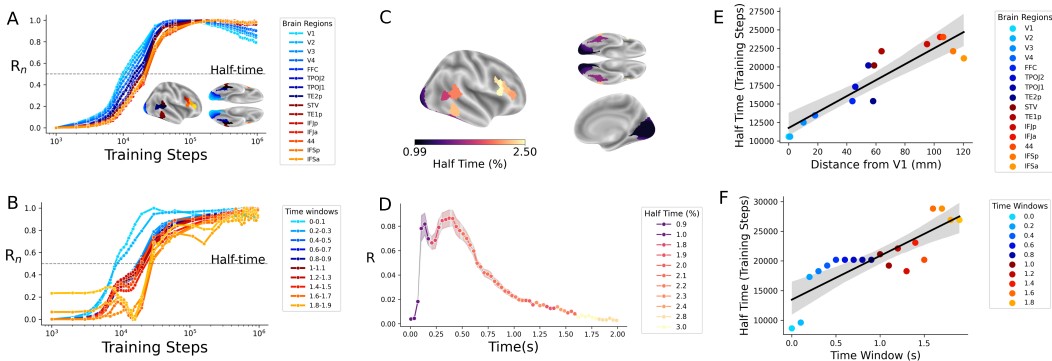

Figure 4: **Emergence of brain-like representations. A.** Normalized brain encoding scores of layer 1 as a function of training for each brain ROI. The dashed line indicates the 50% of the maximum encoding score for each region. **B.** Same as A for MEG time regions of interest (tROIs). **C.** Half time for each brain ROI. **D.** Half time for each time ROI. **E.** Correlation between half time of encoding score across training, and distance of each ROI from V1. **F.** Correlation between half time of encoding score across training, and time position of the encoded cognitive process (tROI).

times than late time windows (e.g. >1,500 ms), Fig 4B,D,F. The correlation between half time and temporal peak is R=0.84, p< $1e^{-5}$.

Overall, these results show that the brain responses of the sensory and prefrontal cortices contain representations of images that are acquired early and late in the training of DINOv3, respectively.

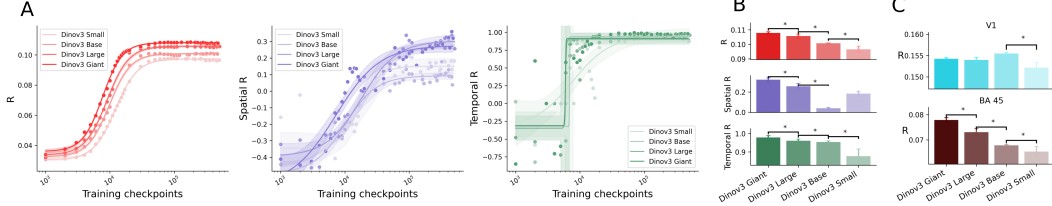

Figure 5: **Impact of model size**. For inter-model comparisons, significance to p< $1e^{-3}$ are represented by asterisks ∗. **A.** Encoding (reds), spatial (purples), and temporal scores (greens) as a function of training and model size. Logarithmic fits of scores across training. **B.** Scores on the final k=$4e^5$ training step. **C.** Encoding scores for V1 and Brodmann area 45 at the end of training.

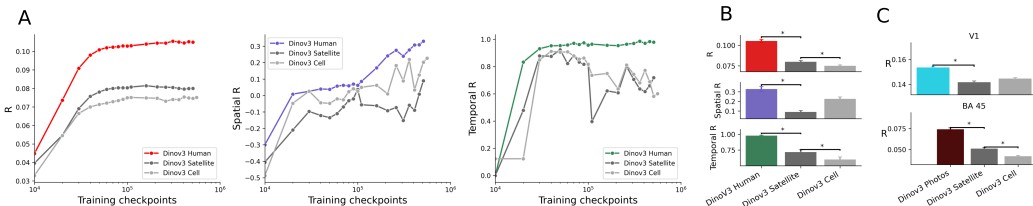

Figure 6: **Impact of image type**. For inter-model comparisons, significance to p< $1e^{-3}$ are represented by asterisks ∗. **A.** Encoding (reds), spatial (purples), and temporal scores (greens) as a function of training and image type. **B.** Scores on the final k=$4e^5$ training step. **C.** Encoding scores for V1 and Brodmann area 45 at the end of training.

**Impact of model size.** How does model size impact convergence? DINO models of larger scale appear to converge quicker and encode higher-level ROIs more accurately. Model size consistently leads to bigger encoding scores at the end of training ($R_{\text{Giant}} = 0.107 > R_{\text{Large}} = 0.105 >$

$R_{\text{Base}} = 0.101 > R_{\text{Small}} = 0.096$ with p $< 1e^{-3}$). Similar, although noisier phenomena can be observed for spatial scores and temporal scores (p $< 1e^{-3}$), Fig 5A, B. Does model size impact encoding scores similarly across ROIs? Applying the same analysis for each ROI separately shows that model size primarily increases encoding of higher-level cortices like BA45, IFS as compared to visual cortices like V1, V2. All models present this size-dependent increased encoding significantly in higher-ROIs, only the smallest ones in V1, V2 (p $< 1e^{-3}$) (Fig 5C, Fig S10 for all studied ROIs).

**Impact of image type.** To assess how image types influence the development of brain-like representations in a model, we train, from scratch, three distinct DINO models, each using one of three natural images datasets: satellite images, cell images and classic (human-centric) images. We focus on a single DINOv3 architecture (Dino Large), with a fixed training length and training data quantity (10M images) and data type as the only varying factor. Training improves encoding scores, spatial scores and temporal scores for all image types (Fig 6A), suggesting that these models learn visual features that are universal across these different types of natural images. However, these brain-similarity metrics are lower for satellite and cell images than for human centric images, for encoding, spatial and temporal scores (Fig 6A, B). Interestingly, this difference is observed across all studied regions of interest: e.g. both V1 (p $< 1e^{-3}$) and BA45 (p $< 1e^{-3}$) are better encoded by a model trained with human centric photos than other models (Fig 6C), see S11 for results on all studied ROIs. At the end of training, DINO human reaches a significantly higher performance regarding encoding, temporal and spatial scores (p $< 1e^{-3}$), Fig 6B. These results might unravel from the fact that human centric images reflect visual input that humans are exposed to, whereas satellite images and cell images are images that human brains have not been trained to process.

## 3.3 LINK TO CORTICAL PROPERTIES

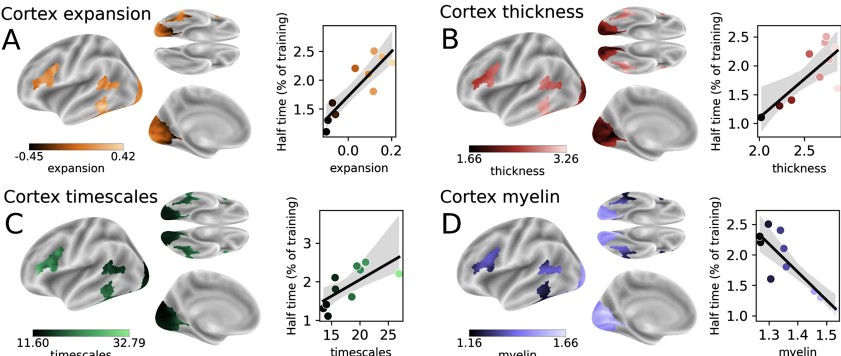

Figure 7: **Relation between shared representations and cortical properties. A.** Left. Cortical expansion index, as estimated from the difference between adults and infants' brains, for each ROI (Hill et al., 2010). Right. Correlation between cortical expansion and half time. Each dot is an ROI. **B.** Same as A for cortical thickness, as estimated from (Van Essen et al., 2013). **C.** Same as A for cortical time scales, as estimated from MEG source reconstruction in (Shafiei et al., 2021). **D.** Same as A for myelin concentration, as estimated from (Van Essen et al., 2013).

Is the development of brain-like representations predicted by functional, structural and developmental properties of the cortex? To explore this issue, we evaluate the correlation between the representational half time of encoding and four properties of the cortex.

**Cortical expansion.** First, we focus on the developmental expansion of cortical regions. Using an atlas comparing infant and adult cortical structures (Hill et al., 2010), we found a strong positive correlation (R=0.88, p $< 1e^{-3}$) between half time and cortical expansion (Fig 7A). This indicates that cortical areas marked by greater developmental growth are also those whose representations emerge later in the AI model.

**Cortical thickness.** Second, we assess the correspondence with cortical thickness, utilizing HCP S1200 estimates. Our results show a significant correlation (R=0.77, p $< 1e^{-2}$), suggesting that cortical areas with larger cortical sheets exhibit longer half times (Fig 7B).

**Cortical dynamics.** Third, the areas with the slowest intrinsic dynamics, as estimated from a source-reconstruction of MEG activity, are also those that tend to have the longest half times (R=0.71, p =

.022). This result directly echoes our MEG results (Fig 2), whereby deeper layers of DINOv3 tend to be associated with slower brain responses (Fig 7C).

**Cortical myelin.** Finally, this dynamic property appears linked to myelin concentration (Van Essen et al., 2013). Myelin, which facilitates faster neuronal transmission, demonstrated a strong negative correlation with half time (R=-0.85, p-val $=1e^{-3}$). This implies that higher myelin concentration is associated with shorter half times (Fig 7D).

In summary, these findings demonstrate a strong predictive relationship between the speed at which brain-like representations emerge in AI models and various structural and functional characteristics of the cortex, across development and once developed.

## 4 DISCUSSION

**Main findings.** Understanding why artificial neural networks develop representations that resemble those in the human brain remains a fundamental challenge to neuroscience and AI (Huh et al., 2024; Hasson et al., 2020; Shen et al., 2025; Caucheteux & King, 2022). While recent studies have documented brain–model similarities across a wide range of architectures and training paradigms (Wang et al., 2023; Conwell et al., 2022), the exact factors that cause this convergence and their interactions remain unclear. Here, we independently manipulate three factors – model size (from DINOv3 small to giant), training length (from 0 to $1e^7$ steps on several training sets of 10M and of 1.7B images) and image type (human-centric, satellite images and biological images) to test how each of them contributes to the emergence of brain-like representations of natural images. Our findings demonstrate that these three factors all independently and interactively impact the extent to which a self-supervised model converges to brain-like visual processing. Results show that size, training duration, and data type each shape the emergence of brain-like representations but also brain-like hierarchies, measured through spatial and temporal scores. Representations, temporal and spatial hierarchies all emerge – though at different pace across training. When comparing untrained and trained versions of seven diversified vision models, we find that the vast majority of them consistently develop qualitatively similar encoding, spatial and temporal scores as DINOv3. Our analyses complement prior work examining only convergence between multiple models (Huh et al., 2024), demonstrating that this convergence also extends onto human neural representations. Finally, we find that the emergence of brain-like properties during training follows the developmental maturation of the human cortex from birth through early adulthood. Although these developmental trajectories have not been tested in other models, the convergence of encoding, spatial, and temporal scores for eight diversified vision models may suggest that these additional results regarding DINOv3 are generalizable to other models. Future research will allow to test this hypothesis.

**Nativism and empiricism.** In particular, the model–brain similarity increases consistently with larger DINOv3 architectures, longer training, and more ecologically valid data. These results are consistent with an increasing set of studies showing linearly aligned representations of natural images (Yamins et al., 2014b; Kriegeskorte, 2015; Schrimpf et al., 2018; Tang et al., 2025; Thobani et al., 2025), with a hierarchy that maps the functional organization of the visual cortices (Eickenberg et al., 2017; La Tour et al., 2022), and dynamics that reflect the ordering of the model's layers (Seeliger et al., 2018; Cichy et al., 2016). In addition to its factorial disentanglement, our study provides additional contributions.

First, this model-brain alignment is not confined to the visual pathways (Eickenberg et al., 2017; Schrimpf et al., 2018; Tang et al., 2025) but extends into high-level – multi-modal – regions of the cortex, including the prefrontal cortex (although see e.g. Solomon et al. (2024) for a low-dimensional set of image features identified in the prefrontal lobe).

Second, our independent manipulation of model size, training duration, and data type further show how these factors *interact* with one another: the largest architectures best align with brain activity as (1) they get trained and (2) on ecologically-relevant naturalistic images.

Third, even non-human-centric datasets (satellite images, biological images) support partial convergence in early visual areas, implying that low-level statistics shared across environments are sufficient to bootstrap early representations. Our findings indicate that models trained on human-centric images still tend to develop representations that more closely resemble those of the human visual system. However, it remains unclear whether this advantage reflects low-level image statistics (e.g., natural color and texture distributions) or higher-level semantic properties typical of hu-

man experience. Additionally, it remains unclear whether this higher alignment for human-centric dataset is driven by a distribution of images more similar to the one in the training data of DINOv3. Distinguishing between these factors will require future research, for example by evaluating brain responses from participants watching controlled non-human centric images. Overall, these results suggest that while the architecture supplies a potential, the data remain critical in making these systems learn representations that are similar to the brain. This interaction between architectures, training and data provides an empirical framework to the long-standing debates in cognitive science on nativism versus empiricism – showing how 'innate' and 'experiential' interact with one another in the development of cognition.

**Towards a model of the visual cortex ontogeny.** This model-brain alignment follows a surprisingly steady developmental trajectory. Early in training, the models rapidly acquire representations that align with the fast and low-level visual responses of the sensory cortices. In contrast, the emergence of slow and high-level representations – particularly those aligning with the prefrontal cortex – appears to require both far more training data.

This developmental trajectory echoes the biological development of the human cortex: the brain areas with which the AI models align last during their training are precisely those with the greatest cortical thickness, slower intrinsic timescales, prolonged maturation, and lower levels of myelination – i.e. the areas of the associative cortices that are known to slowly develop throughout the first two decades of life (Dehaene, 2021). This result suggests that the sequential acquisition of representations in artificial neural networks may spontaneously model some of the developmental trajectories of brain functions. In doing so, they may ultimately provide a new computational framework to understand the staged maturation of visual processing in biological systems (Vogelsang et al., 2024; Zaadnoordijk et al., 2022; Long et al., 2024).

**Open questions.** Several results were not anticipated. First, the temporal score, encoding score and spatial score do not appear to emerge simultaneously – hence leading to the novel question of why these metrics follow this specific order. The factors that lead the temporal score to emerge are not entirely clear. However, the temporal score rises before the encoding score 3, suggesting that the encoding score alone does not solely explain the temporal score. Second, the spatial and temporal scores are initially negative (respectively significantly, p = 0.05, and non-significantly, p > 0.05) at the beginning of model training.This means that the deepest layers of a random DINOv3 tend to best predict fast and low-level brain responses at the very early (but not late) stages of training. Finally, the half times of these three metrics are reached in between 1% and 4% – i.e. only n=1.6B images – of DINOv3 training quantity. This suggests that while low-level brain-like representations are very quickly learnable, the high-level representations of the brain require a very large amount of data to be fully acquired.

**Limitations.** While this study offers a controlled analysis of brain–model convergence, several limitations warrant consideration. First, our findings are based exclusively on a single family of self-supervised vision models (DINOv3), which are hierarchical by design. It thus remains an open question whether similar spatial, temporal and encoding scores would emerge with other architectures and training objectives (Conwell et al., 2021). Second, fMRI and MEG offer limited resolution and thus provide coarse population-level brain activity and may overlook fine-grained neural mechanisms. Third, our analyses focus solely on the adult brain, leaving open the question of how these alignments emerge across development. Understanding when these correspondences arise will require data from infants, children, or longitudinal cohorts (Evanson et al., 2025). Additionally, we analyze two datasets where participants watched images most often passively: future work should assess how different tasks modulate the alignment between DINOv3 and the prefrontal cortex. Future research should also extend this systematic exploration to additional factors driving brain-like convergence, as well as to the interactive effects between all of them. Finally, while we quantify the similarity between representations from models and the brain, the exact nature and semantic structure of these neuronal representations continues to be a subject of intense ongoing research (Gifford et al., 2025; Graumann et al., 2022). Closing this interpretability gap certainly remains a major challenge to both neuroscience and AI.

**Conclusion.** Beyond the characterization of the spontaneous convergence between AI models and brains, these findings chart a path toward using AI models as tools to investigate the organizing principles of biological vision in the human brain. By showing how machines can come to see like us, our findings provide cues as to how the human brain may come to see the world.

## A    REPRODUCIBILITY STATEMENT

The paper is based on an open-source MEG dataset (Hebart et al., 2023a) and an open-source fMRI dataset (Allen et al., 2022), as well as several open-source vision models (versions of DINOv3 (Siméoni et al., 2025)). These models and datasets are cited in the Introduction and the Methods sections. Regarding the comparative trainings, the satellite and human-centric datasets are currently proprietary, whereas the cellular dataset is a collection of open-source datasets (Lorenci et al., 2025). Linear decoding code is available at:

`https://mne.tools/stable/generated/mne.decoding.SlidingEstimator.html`.

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

# B APPENDIX

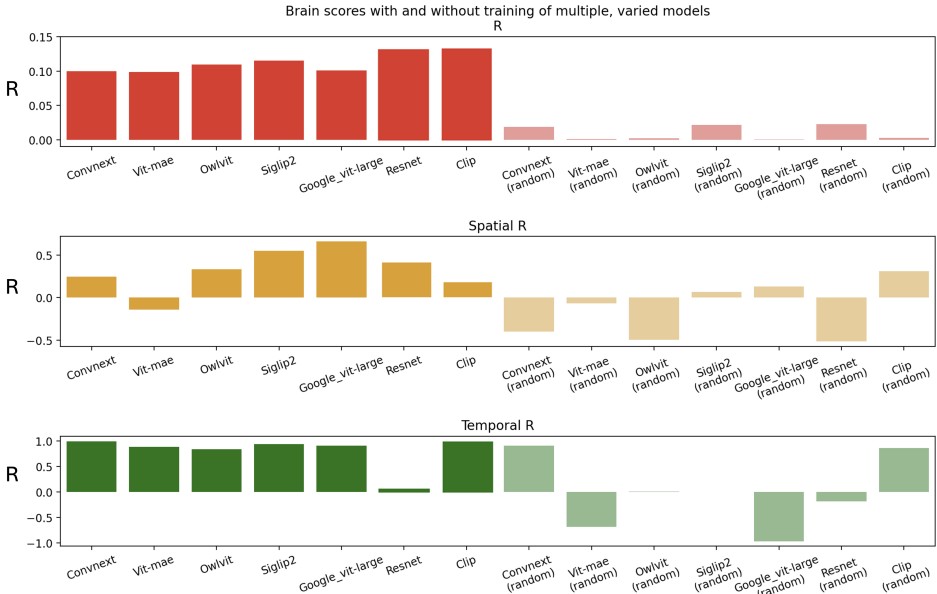

Figure S1: **Encoding, spatial and temporal scores reproduced for a range of seven vision models** with varying architectures and training objectives - including CNNs (ResNet-50, ConvNeXt-Large), a self-supervised ViT with different objective than DINOv3 (ViT-MAE, masked image reconstruction), a supervised ViT (ViT-L/16), and vision–language contrastive transformers (CLIP, SigLIP2, OWL-ViT).

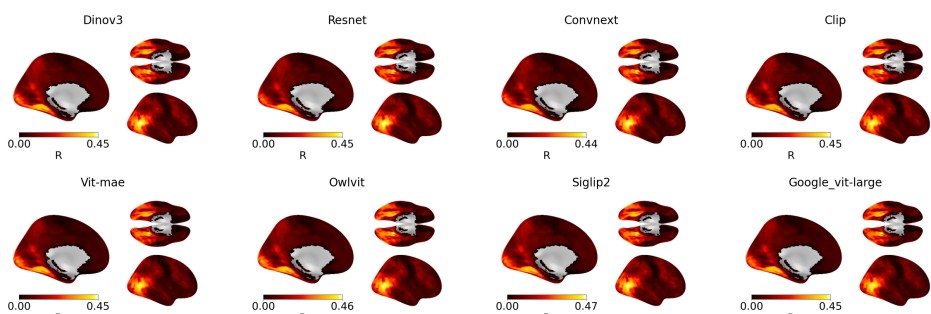

Figure S2: **Encoding scores across cortex, reproduced for a range of seven vision models** with varying architectures and training objectives – including CNNs (ResNet-50, ConvNeXt-Large), a self-supervised ViT with different objective than DINOv3 (ViT-MAE, masked image reconstruction), a supervised ViT (ViT-L/16), and vision–language contrastive transformers (CLIP, SigLIP2, OWL-ViT).

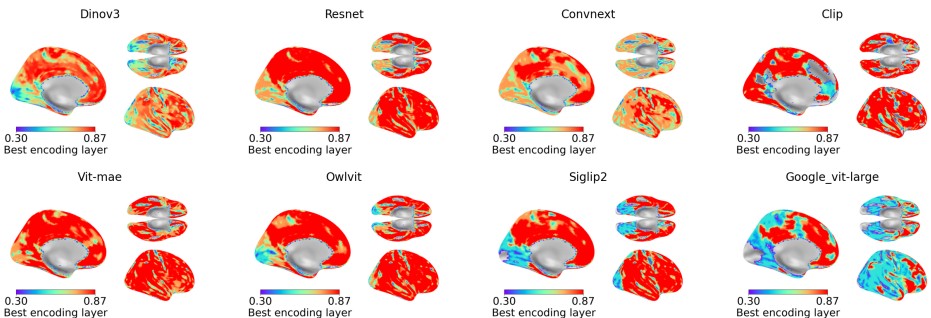

Figure S3: **Maximally encoding layers across cortex, reproduced for a range of seven vision models** with varying architectures and training objectives – including CNNs (ResNet-50, ConvNeXt-Large), a self-supervised ViT with different objective than DINOv3 (ViT-MAE, masked image reconstruction), a supervised ViT (ViT-L/16), and vision–language contrastive transformers (CLIP, SigLIP2, OWL-ViT).

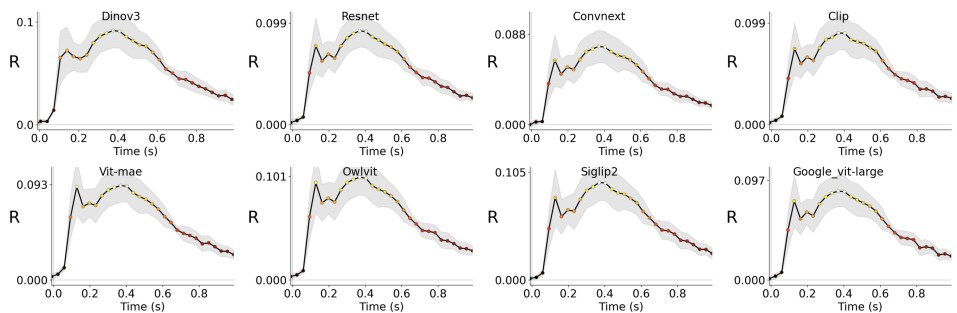

Figure S4: **Encoding scores along time, reproduced for a range of seven vision models** with varying architectures and training objectives – including CNNs (ResNet-50, ConvNeXt-Large), a self-supervised ViT with different objective than DINOv3 (ViT-MAE, masked image reconstruction), a supervised ViT (ViT-L/16), and vision–language contrastive transformers (CLIP, SigLIP2, OWL-ViT).

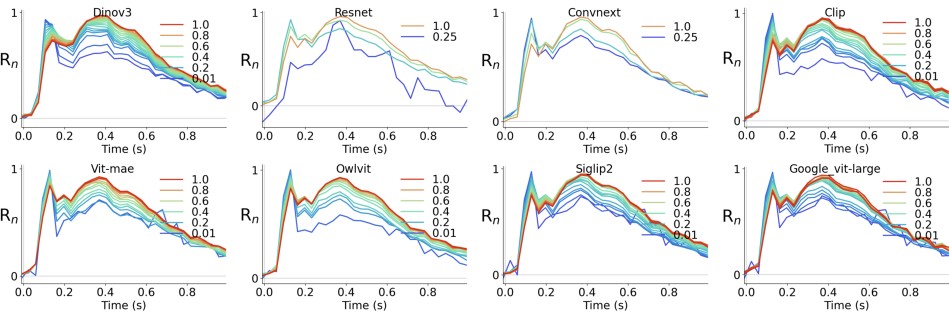

Figure S5: **Encoding scores for each layer along time, reproduced for a range of seven vision models** with varying architectures and training objectives - including CNNs (ResNet-50, ConvNeXt-Large), a self-supervised ViT with different objective than DINOv3 (ViT-MAE, masked image reconstruction), a supervised ViT (ViT-L/16), and vision–language contrastive transformers (CLIP, SigLIP2, OWL-ViT).

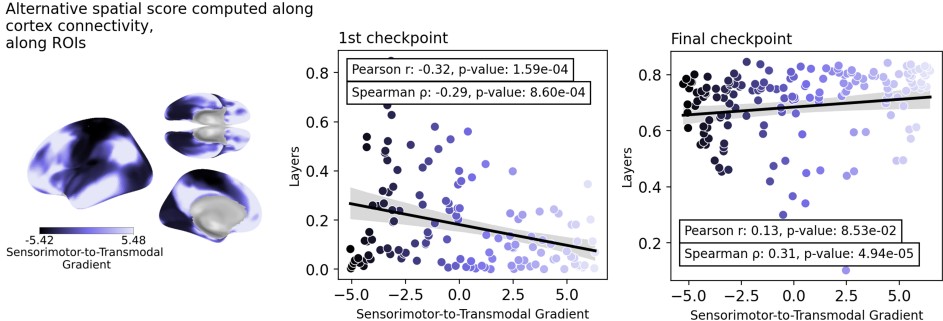

Figure S6: **Alternative spatial score along the sensory-to-transmodal gradient**. Partial reproduction of the spatial scores results obtained on DINOv3 using the value of each ROI along the sensory-to-transmodal gradient map (Margulies et al., 2016) instead of the euclidean distance of this ROI from V1. We obtain a similar increase across training from -0.32 to 0.13 (1st to last checkpoint).

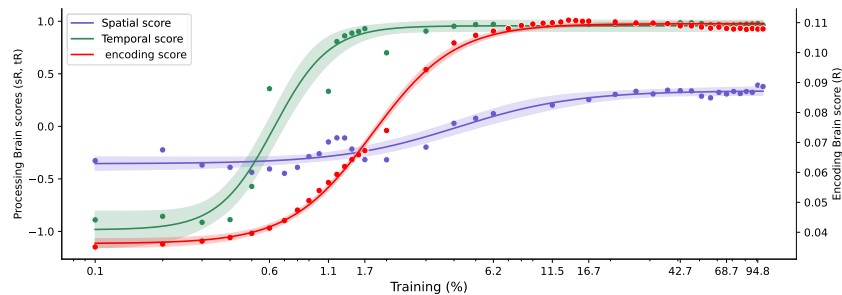

Figure S7: **Evolution of temporal, encoding and spatial scores** as a function of DINOv3's training.

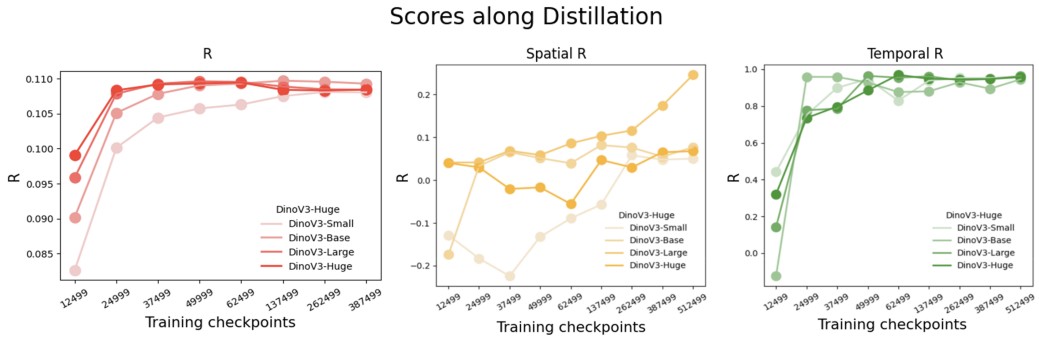

Figure S8: **Impact of model size along distillation for DINOv3-Small, Base, Large and Huge**. Encoding (reds), spatial (yellow), and temporal scores (greens) as a function of training and model size.

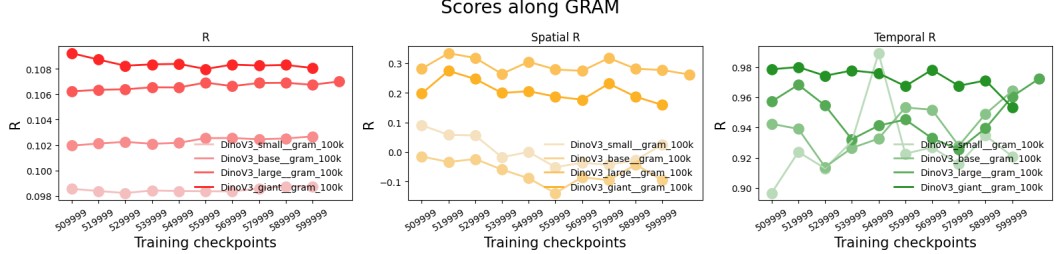

Figure S9: **Impact of model size along GRAM anchoring for DINOv3-Small, Base, Large and Huge**. Encoding (reds), spatial (yellow), and temporal scores (greens) as a function of GRAM anchoring and model size, for models trained from scratch.

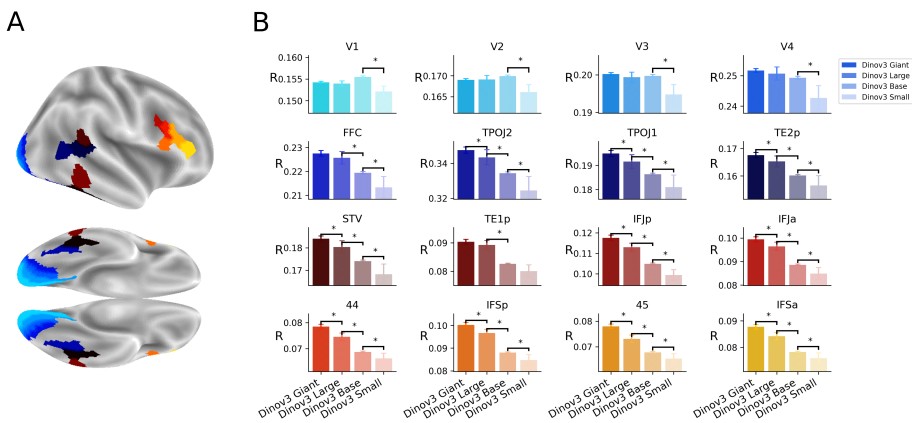

Figure S10: **Impact of model size for multiple ROIs.** For inter-model comparisons, significance to p$< 1e^{-3}$ are represented by asterisks $*$. **A.** Brain ROIs. **B.** Encoding scores for each ROI at the end of training.

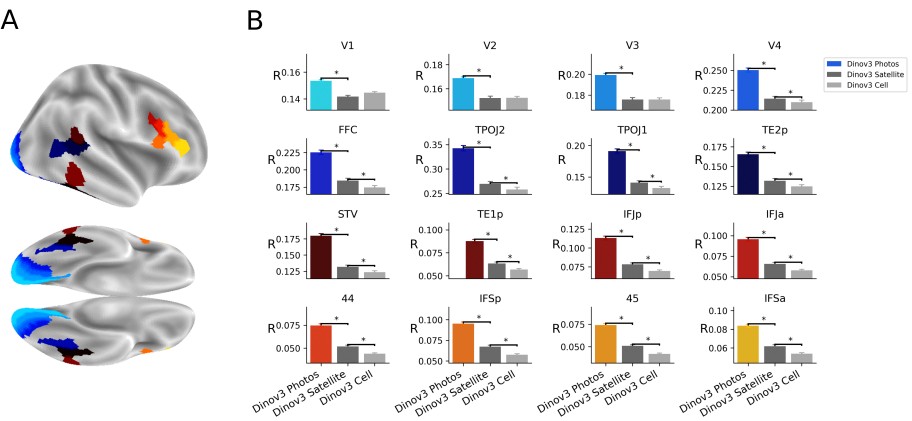

Figure S11: **Impact of image type for multiple ROIs.** For inter-model comparisons, significance to p$< 1e^{-3}$ are represented by asterisks $*$. **A.** Brain ROIs. **B.** Encoding scores for each ROI at the end of training.

## C  Ethics Statement

Among the three datasets this study leverages, only the human-centric dataset is constituted from images from the web and imageNet; each of these images went through platform-level content moderation to prevent any harmful contents (Siméoni et al., 2025).
The present paper uses two datasets regarding research with human subjects, both these datasets have already been published, are open-source and cited in the paper. These open-source recordings were collected after participants' informed consent and were validated by the corresponding Institutional Review Boards.

## D  Acknowledgments

We warmly thank the authors of (Hebart et al., 2023a) and (Allen et al., 2022) for building and open-sourcing the MEG and fMRI datasets on which we relied for this study. We thank the Meta communication and legal teams for their support throughout the submission process.

## E  LLM use

An LLM was used in preparing this manuscript for proofreading purposes.

