# B APPENDIX

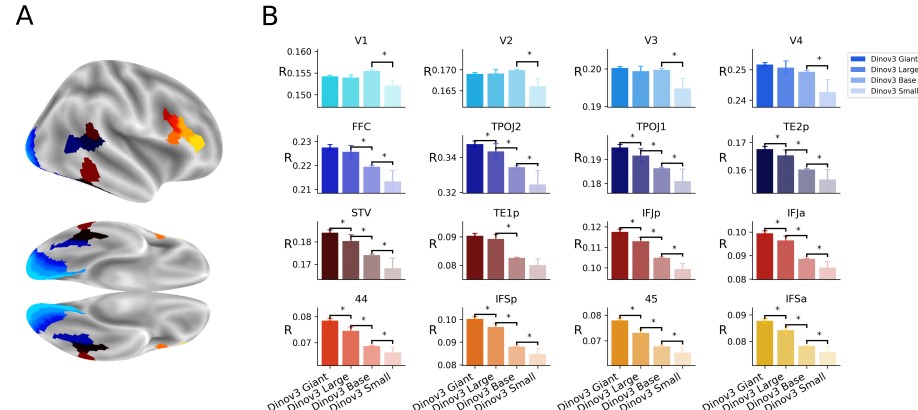

Figure S1: **Impact of model size for multiple ROIs.** For inter-model comparisons, significance to p$< 1e^{-3}$ are represented by asterisks $*$. **A.** Brain ROIs. **B.** Encoding scores for each ROI at the end of training.

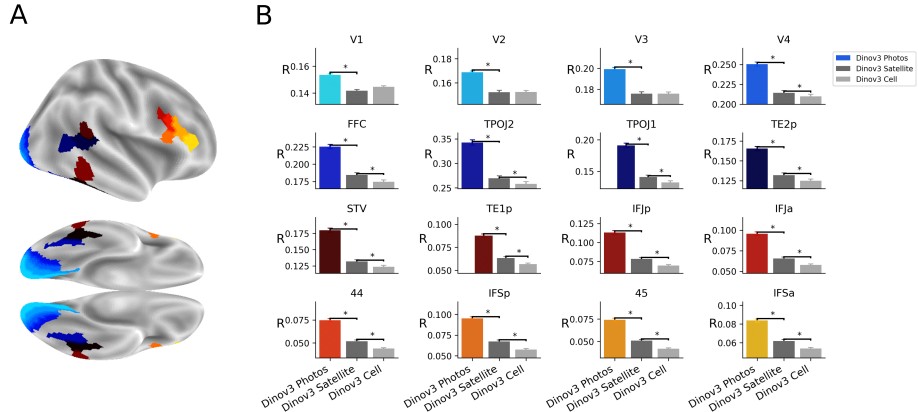

Figure S2: **Impact of image type for multiple ROIs.** For inter-model comparisons, significance to p$< 1e^{-3}$ are represented by asterisks $*$. **A.** Brain ROIs. **B.** Encoding scores for each ROI at the end of training.