# OpenReview forum: "Disentangling the Factors of Convergence between Brains and DINOv3"
_ICLR.cc/2026/Conference — ICLR 2026 Poster_

### Official Review · Reviewer_omL2 · 2025-10-24

**Soundness:** 4
**Presentation:** 3
**Contribution:** 4
**Rating:** 8
**Confidence:** 4

**Summary:**

The present paper studies the factors that drive brain-model alignment. In particular, it investigates three dimensions: model size, training time, and training data. The authors train DINOv3 from scratch and compute encoding, spatial, and temporal scores. The three varied dimensions interactively impact each of these alignment metrics: (1) the largest models reaches the highest alignment, (2) the emergence of brain-like representations follows a specific chronology during training, and (3) the developmental trajectory is indexed by both structural and functional properties of the human cortex.

**Strengths:**

I found this paper to be very insightful and interesting to read. The systematic analysis of differnet model properties on brain alignment is one of the most important analyses to be done in this field.

The methods and data are very sound (although I cannot speak a some of the very specific MEG/fMRI details). The paper builds on a wealth of prior work, but complements it nicely by taking an important next step. It was easy to read and follow, and is a great thematic fit for ICLR.

In particular, I found the analyses regarding the temporal and spatial score, as well as the alignment trajectories during training, to be very interesting -- I haven't seen something like this before.

**Weaknesses:**

While the authors vary three different dimensions, they don't explore the whole space (e.g. they did not look at varying the data in all model sizes). This risks of missing potential interaction effects.

The present study considers only a single model class. This is not a major problem from my side, and it is already acknowledged and discussed by the authors.

Minor: one could use consistent typesetting for R (italic or not).

**Questions:**

Personally, I found the analyses regarding the training trajectories of the alignment metrics to be the most interesting part. Has there any work been done on this before already? That would help to judge the novelty of the present approach.

Are images from THINGS and the Natural Scenes Dataset in the Human centered 1.7B data? If so, does this matter for the presented results?

For Figure 3: the temporal score is only based on the MEG data, the spatial score soley on fMRI data, and the encoding score on both, right? Based on this, are these three metrics directly comparable to each other? The fact that that they are in the same figure and the wording in the text seem to suggest so, but I am a bit unsure.

---

> ### Author Response · Authors · 2025-11-24
> **Generalization to other vision models: We now add 10 additional sets of results (among which a benchmark of 7 new models) and modify the manuscript**
>
> We thank Reviewer omL224 for their thorough review.
>
> We now add 10 additional sets of results  (among which a benchmark of 7 new models) and amend sections of the manuscript to address the issues raised and improve the paper. We uploaded the updated version of the paper, with modified text and the new figures.
>
> **1. Discussion on Dinov3 vs generalization of the results to other trained and untrained vision models (self-supervised, supervised ViTs, vision–language contrastive transformers, CNNs)**
>
> The study was previously conducted specifically on DINOv3. To reflect this comment, we reproduced the results and 7 other diversified vision models and modified the manuscript.
>
> **1A. Clarification of the contribution**
>
> First, we modified the title, intro, abstract and discussion of the paper to highlight the focus on DINOv3, as follows:
> - Title: To reflect the focus on DINOv3 and the work along DINOv3’s training, we modify the title to “Disentangling the Factors of Convergence between Brains and DINOv3”
> - Abstract: “We trained a family of self-supervised DINOv3 vision transformers”
> - Introduction: “we systematically train a variety of DINOv3 models”
> - Discussion: “The model–brain similarity increases consistently with larger DINOv3 architectures”
>
> **1B. Reproduction of key results on seven vision models - With trained models and untrained controls**
>
> Additionally, we reproduced the results of the manuscript across seven architectures and training objectives – including CNNs (ResNet-50, ConvNeXt-Large), a self-supervised ViT with different objective than DINOv3 (ViT-MAE, masked image reconstruction), a supervised ViT (ViT-L/16), and vision – language contrastive transformers (CLIP, SigLIP2, OWL-ViT).
>
> We add scores obtained with random-weight models as a control. These analyses compare brain score, spatial score, temporal score in untrained as well as fully trained versions of all these seven models.
>
> Results show that once trained:
> - The vast majority of vision models consistently have qualitatively similar encoding, spatial and temporal scores.
> - The trained vision models consistently have higher encoding, spatial and temporal scores than their untrained counterparts. These scores also improve across the training of DINOv3 in Fig. 3 of the manuscript.
> - The trained models tend to have similar encoding, spatial and temporal scores between models.
> - The scores of untrained vision models tend to vary more between models. A hypothesis for this behavior is that untrained models do not share the same inductive biases and start from very different initial points – yet they all eventually converge to similar representations and computational pathways, measurable through  encoding, spatial and temporal scores.
>
> To report these additional analyses, we add five figures S1, S2, S3, S4 and S5 in Appendix, presenting a comparative analysis of trained vs untrained versions of the 7 models studied, as well as brainplots and curves along time of encoding scores and maximally encoding layers.
>
> We adapted the Results sections to these new results as follows:
> 3.3.1 DINOv3-Brain similarity
> “To test for generalization of Encoding, Spatial and Temporal scores we reproduce these results on a variety of seven vision models including CNNs, supervised and self-supervised ViTs as well as vision-language contrastive transformers. We find similar scores for all three metrics, across all these models.”
>
> 3.3.2 What factors lead DINOv3 to become brain-like?
> “Across seven vision models with diverse architectures – CNNs, supervised ViTs and vision-language contrastive transformers –, trained models consistently show higher  encoding, spatial, and temporal scores than untrained models. Additionally, trained models tend to show convergent encoding, spatial, and temporal scores, whereas scores from untrained models vary more widely – likely reflecting differences in their inductive biases. These additional analyses are reported in Figs. S1–S5, comparing trained and untrained versions of all models.”
>
> **1C. Reasons for focusing on DINOv3 for analyses along trainings**
>
> We chose to study exclusively DINOv3 for 3 reasons:
>
> - It was the only model for which we could access checkpoints across the whole training from scratch.
> - We needed a common architecture for which to vary only one single parameter for each training (size, data type, etc.) to be able to disentangle factors of convergence with representations of the brain.
> - DINOv3 was the only model that we had the possibility to train multiple times from scratch, using controlled variants of the model and training on controlled sets of data from different types (cellular, satellite, human-centric). Self-supervision is also well suited regarding training on different types of data.

---

> ### Author Response · Authors · 2025-11-24
> **Interaction between factors, Data distribution, Metric Comparability, and Novel Contributions**
>
> **2. Interaction between factors**
>
> We did not systematically vary training data across all model sizes. However, we already examined pairwise interactions between factors (e.g., size × training, data type × training), which showed that small models trained longer still perform worse than larger models trained for fewer steps. Investigating the full three-way interaction would be computationally costly, and in practice smaller or larger models often failed to converge on non–human-centric datasets.
>
> This is a good point to try to reach, and we have therefore added a statement in the Discussion regarding the interaction between model size and data type:
>
> “Future research should also extend this systematic exploration to additional factors driving brain-like convergence, as well as to the interactive effects between all of them.”
>
> **3. Images of the Human centered 1.7B data**
>
> This is indeed an interesting point. We cannot check if the same images are present in both datasets, however even if there were not, the distribution of images may indeed be much more similar between the dataset used for DINOv3’s training and the human-centric dataset, than with the cellular and satellite datasets.
>
> We now state this as a limitation in the discussion: “it remains unclear whether this higher alignment for human-centric dataset is driven by a distribution of images more similar to the one in the training data of DINOv3. Distinguishing between these factors will require future research, for example by evaluating brain responses from participants watching controlled non-human centric images.
>
> **4. Are these three metrics directly comparable to each other?**
>
> Indeed, the temporal and spatial scores come from different recording modalities (fMRI for spatial resolution and MEG for temporal resolution), but all three metrics are derived from the same underlying encoding framework. Each captures a different form of convergence through Pearson score – representation, spatial hierarchy, or temporal hierarchy – and their emergence during training, as well as their dependence on data type and model size, can therefore be meaningfully compared.
>
> **5. Discussion on novelty regarding trajectories of the alignment metrics during training, and summing up the contributions of the paper**
>
> Thank you for this comment. To the best of our knowledge, we are indeed the first study to report trajectories of several alignment metrics during training. Caucheteux & King, (2022) did report evolution of encoding score through training of models but in the language domain, not vision, and with no other alignment metrics, also without disentangling of factors allowing this convergence. Wang et al (2023) and Conwell et al. (2021) compared different vision models though not along training, with no other alignment metrics regarding hierarchy (temporal and spatial scores), also without disentangling of factors allowing this convergence.
>
> - Caucheteux & King (2022). Communications Biology.
> - Conwellet al. (2021). SVRHM 2021 Workshop @ NeurIPS.
> - Wang et al. (2023). Nature Machine Intelligence.
>
> To clarify and sum up the novel contributions of the paper:
> - **Controlled training**: We disentangle co-varying factors such as training data type, dataset size, and architectural depth.
> - **New similarity measure for hierarchical alignment**: We measure, beyond representations, similarities in hierarchies of processing – i.e. temporal and spatial scores.
> - **Analysis along training checkpoints**: We study how these brain-like properties emerge over the course of training. By disentangling these factors and showing, for example, that spatial alignment emerges earlier than temporal alignment, we provide new evidence that clarifies and extends the mechanisms underlying model–brain convergence.
> - **Cortical markers**: We find that the emergence of brain-like properties during training partially mirrors the developmental maturation of the human cortex from birth through early adulthood.
>
> We now made these novel contributions more explicit in the discussion:
>
> “This study systematically disentangles co-varying factors such as training data type, dataset size, and architectural depth. Results show that size, training duration, and data type each shape the emergence of brain-like representations but also brain-like hierarchies, measured through spatial and temporal scores. Representations, temporal and spatial hierarchies all emerge - though at different pace across training. Our analyses complement prior work examining only model–model convergence (Huth et al, 2024), demonstrating that this convergence also extends onto human neural representations. Finally, we find that the emergence of brain-like properties during training follows the developmental maturation of the human cortex from birth through early adulthood.”
>
> **6. Consistent typesetting for R**
> Thank you for this good catch, we now fixed it.

---

### Official Review · Reviewer_9Mpq · 2025-10-31

**Soundness:** 3
**Presentation:** 3
**Contribution:** 3
**Rating:** 6
**Confidence:** 3

**Summary:**

The authors train multiple different DINOv3 models with varying model sizes (s, B, L, g, 7B) and different pretraining datasets (Human centered, Satellite, Cellular). Given checkpoints of these models at different training iterations, they compare the representations with the representations of the human brain recorded with fMRI and MEG on the THINGs and natural scenes datasets. The study finds that (1) larger models converge quicker and encode higher-level ROIs more accurately, (2) longer training and (3) more human-like data (in contrast to medical/satellite data) leads to a higher degree of model–brain similarity.

**Strengths:**

* The paper studies an interesting question and the methodological comparisons between the DINOv3 models and brain measurements is sound (2 different measurements, independent datasets, 3 different metrics).

* The paper is well written and explains the methods and results in an understandable way.

* The alignment is studied over many different training iterations, allowing insights into how it progresses over the training time which is particularly interesting and often missing in other similar studies.

* The authors not only evaluated the representational alignment but also the spatial correlation between the location in the brain and the different layers in the ViT giving insights whether the representations are constructed/progress in a similar way.

**Weaknesses:**

* The study only investigates DINOv3 models and does not investigate any other SSL objective or image/text, supervised objectives. The paper is called: “Disentangling the Factors of Convergence between Brains and Computer Vision Models” but only looks at different training iterations (#samples seen), model size and dataset type (human centered, satellite, cells). Model architecture and training objectives are not studied. In previous studies [1], the objective was a key driver of alignment to human similarity judgements. Therefore, it would be pretty interesting to study how the score differs in relation to publicly available image/text models, supervised models, SSL models that only rely on masking (e.g. MAE [2], CAPI [3], AIMM [4]) and the ones relying on contrastive/multi-view objectives (SimCLR [5], MoCo [6], BYOL [7]). This is partly acknowledged in the limitations section but the title and some of the claims are too general (speaking of Computer Vision Models) given that these factors are not investigated.

* Isn’t the “Impact of image type” analysis confounded by the fact that the datasets that were used to gather the fMRI, MEG data are also more human-like (THINGs and natural scenes)? I would be curious to see how that changes when the alignment is evaluated on medical/cell images and satellite images. This should be discussed in this section.

* Minor typos: In line 239 a space is missing. In line 301, there is one space too much before “?”.


[1] Muttenthaler, Lukas, et al. "Human alignment of neural network representations." ICLR 2023.

[2] He, Kaiming, et al. "Masked autoencoders are scalable vision learners." Proceedings of the IEEE/CVF conference on computer vision and pattern recognition. 2022.

[3] Darcet, Timothée, et al. "Cluster and predict latent patches for improved masked image modeling." TMLR 2025.

[4] El-Nouby, Alaaeldin, et al. "Scalable pre-training of large autoregressive image models." ICML 2024.

[5] Chen, Ting, et al. "A simple framework for contrastive learning of visual representations." International conference on machine learning. ICML 2020.

[6] He, Kaiming, et al. "Momentum contrast for unsupervised visual representation learning." Proceedings of the IEEE/CVF conference on computer vision and pattern recognition. 2020.

[7] Grill, Jean-Bastien, et al. "Bootstrap your own latent-a new approach to self-supervised learning." Advances in neural information processing systems 33 (2020): 21271-21284.

**Questions:**

* The brain “alignment” metrics are shown over the training time, but the DINOv3 framework has several mid-training/posttraining phases (gram anchoring, high resolution training). Did you investigate the effects of this training phases wrt. to the brain alignment scores?

* It was observed that larger models have generally larger correlations to the human brain representations than smaller models. Did you also evaluate the distilled models from the DINOV3 released models? I would be curious whether these would also be able to match the higher prediction scores when having a larger teacher.

* How did you exactly extract the representations at different layers of the model? Were the CLS token, patch tokens, register tokens used and how were they potentially aggregated?

* Did you investigate any more sophisticated probes than linear probes (e.g. attention probes)?

---

> ### Author Response · Authors · 2025-11-24
> **Generalization to other vision models: We now add 10 additional sets of results (among which a benchmark of 7 new models) and modify the manuscript**
>
> We thank Reviewer 9Mpq for their thorough review.
> We now add 10 additional sets of results  (among which a benchmark of 7 new models) and amend sections of the manuscript to address the issues raised and improve the paper. We uploaded the updated version of the paper, with modified text and the new figures.
>
>
> **1. Discussion on Dinov3 vs generalization of the results to other trained and untrained vision models (self-supervised, supervised ViTs, vision–language contrastive transformers, CNNs)**
>
> This is indeed a very good point, the study was conducted specifically on DINOv3. To reflect this comment, we reproduced the results and 7 other diversified vision models and modified the manuscript.
>
> **1A. Clarification of the contribution**
>
>  First, we modified the title, intro, abstract and discussion of the paper to highlight the focus on DINOv3, as follows:
> - Title: To reflect the focus on DINOv3 and the work along DINOv3’s training, we modify the title to “Disentangling the Factors of Convergence between Brains and DINOv3”
> - Abstract: “We trained a family of self-supervised DINOv3 vision transformers”
> - Introduction: “we systematically train a variety of DINOv3 models”
> - Discussion: “The model–brain similarity increases consistently with larger DINOv3 architectures”
>
> **1B. Reproduction of key results on seven vision models - With trained models and untrained controls**
>
> Additionally, we reproduced the results of the manuscript across seven architectures and training objectives – including CNNs (ResNet-50, ConvNeXt-Large), a self-supervised ViT with different objective than DINOv3 (ViT-MAE, masked image reconstruction), a supervised ViT (ViT-L/16), and vision – language contrastive transformers (CLIP, SigLIP2, OWL-ViT).
>
> We add scores obtained with the random-weight models as a control. These analyses compare brain score, spatial score, temporal score in untrained as well as fully trained versions of all these seven models.
>
> Results show that once trained:
>
> - The vast majority of vision models consistently have qualitatively similar encoding, spatial and temporal scores.
> - The trained vision models consistently have higher encoding, spatial and temporal scores than their untrained counterparts. These scores also improve across the training of DINOv3 in Fig. 3 of the manuscript.
> - The trained models tend to have similar encoding, spatial and temporal scores between models.
> - The scores of untrained vision models tend to vary more between models. A hypothesis for this behavior is that untrained models do not share the same inductive biases and start from very different initial points – yet they all eventually converge to similar representations and computational pathways, measurable through  encoding, spatial and temporal scores.
>
> To report these additional analyses, we add five figures S1, S2, S3, S4 and S5 in Appendix, presenting a comparative analysis of trained vs untrained versions of the 7 models studied, as well as brainplots and curves along time of encoding scores and maximally encoding layers.
>
> We adapted the Results sections to these new results as follows:
>
> 3.3.1 DINOv3-Brain similarity
>
> “To test for generalization of Encoding, Spatial and Temporal scores we reproduce these results on a variety of seven vision models including CNNs, supervised and self-supervised ViTs as well as vision-language contrastive transformers. We find similar scores for all three metrics, across all these models.”
>
> 3.3.2 What factors lead DINOv3 to become brain-like?
> “Across seven vision models with diverse architectures – CNNs, supervised ViTs and vision-language contrastive transformers –, trained models consistently show higher  encoding, spatial, and temporal scores than untrained models. Additionally, trained models tend to show convergent encoding, spatial, and temporal scores, whereas scores from untrained models vary more widely – likely reflecting differences in their inductive biases. These additional analyses are reported in Figs. S1–S5, comparing trained and untrained versions of all models.”
>
> **1C. Reasons for focusing on DINOv3 for analyses along trainings**
>
> We chose to study exclusively DINOv3 for 3 reasons:
> - It was the only model for which we could access checkpoints across the whole training from scratch.
> - We needed a common architecture for which to vary only one single parameter for each training (size, data type, etc.) to be able to disentangle factors of convergence with representations of the brain.
> - DINOv3 was the only model that we had the possibility to train multiple times from scratch, using controlled variants of the model and training on controlled sets of data from different types (cellular, satellite, human-centric). Self-supervision is also well suited regarding training on different types of data.

---

> ### Author Response · Authors · 2025-11-24
> **Additional Analyses: Human-Centric Data, GRAM Anchoring, Distillation, Representation Extraction, and Linear Probes**
>
> **2. Discussion on the analysis of “human-centric data”**
>
> This is a very good remark, we modified the discussion to extend on this point:
>
> “Our findings indicate that models trained on human-centric images tend to develop representations that more closely resemble those of the human visual system. However, it remains unclear whether this advantage reflects low-level image statistics (e.g., natural color and texture distributions) or higher-level semantic properties typical of human experience. Distinguishing between these factors will require future research, for example by evaluating brain responses from participants watching controlled non-human centric images.”
>
>  **3. Investigation of the effects of gram anchoring with respect to to the brain alignment scores**
>
> To investigate this point, we computed the evolution of metrics along checkpoints of GRAM anchoring. These stay sensibly constant after the end of main training, along GRAM anchoring, we report these results in Supp. Fig. S9 of the updated manuscript.
>
>
>  **4. Impact of distillation compared to training from scratch on multiple sizes of models**
>
> This is indeed an interesting remark. We now added new distillation experiments. The results show that scores evolve similarly across model sizes and along training than when models are trained from scratch. We report these results in Supp. Fig. S8 of the updated manuscript. As brain development may be closer to "learning from scratch", rather than from distillation, we decided to keep the results of models trained from scratch as the main results, though the distillation results complete the paper.
>
> **5. Modalities of extraction of the representations at different layers of the model**
>
> To extract the representations, we take the average of patch tokens – we made this decision after comparing with the same analyses using the CLS token, which gave a poorer encoding score. We do not use register tokens as they play the part of cleaners of the representations of images to improve training, and are generally less prone to be used as representations of an entire image (Darcet et al., 2024).
>
> Darcet, T., Oquab, M., Mairal, J., & Bojanowski, P. (2024). Vision transformers need registers. In Proceedings of the International Conference on Learning Representations (ICLR 2024).
>
>
> **6. Discussion on use of linear probes**
>
> Thank you for this interesting point, we decided to focus on linear probes to be able to maintain the geometry of latent representations of DINOv3 that we compare to the brain recordings. We now modified the Method paragraph of our manuscript to express this point.
>
> **7. Minor prose issues:**
>
> Thank you for these good catches, we now fixed them.

---

### Official Review · Reviewer_bhwe · 2025-11-03

**Soundness:** 3
**Presentation:** 3
**Contribution:** 2
**Rating:** 4
**Confidence:** 4

**Summary:**

The paper investigates factors contributing to neural network-brain representation alignment by training variants of the self-supervised vision transformer DINOv3 with systematically varied (1) model size, (2) training amount, and (3) image type. Model representations are compared to human brain activity measured via fMRI and MEG using three complementary metrics: encoding score (overall linear similarity), spatial score (hierarchical organization across brain regions), and temporal score. The authors find that larger models trained longer on human-centric images achieve higher brain-similarity scores across all three metrics, and that different brain regions align at different training stages i.e. early sensory regions align first, followed by higher-level prefrontal regions.

**Strengths:**

- The combination of fMRI and MEG with three complementary similarity metrics (overall encoding, spatial, temporal) provides a more complete characterization than prior studies relying on single modalities or metrics, which allows evaluation of alignment across both spatial mappings (cortical hierarchy) and temporal processing sequences rather than using only pre-selected layer-region pairs or fixed time windows (unlike most existing studies).
 -The analysis extends beyond the traditional ventral visual stream to include prefrontal regions (BA44, BA45, IFSa, IFSp),  show linear predictability in higher-level associative cortices, though interpretation requires careful consideration (see weaknesses part)
- The correlational analysis linking alignment emergence speed to cortical properties (developmental expansion, thickness, myelination, intrinsic timescales) provides biological context, though this constitutes post-hoc analysis using existing atlases rather than experimental validation

**Weaknesses:**

- Limited scope (question of generalizability) -
All experiments use DINOv3, It's unclear whether findings generalize to: 1) architecture family with different inductive biases such as CNNs (with presumably much stronger ventral-stream-hierarchical-aligning inductive biases) 2) supervised models (what’s the role of learning objective under the context of diets needed for training?) or (3) other self-supervised methods (e.g., MAE, SimCLR). one could expect conclusions (especially regarding the training dynamics/ relative convergence speed (in terms of half-time) between the three measures) changes accordingly.
- Ambiguity in ROI-level encoding scores
Section 2.2 states scores can be "summarized as the average R score across brain dimensions" but doesn't specify whether ROI-level comparisons (Figures 5C, 6C) use: A) averaging across all layer-region pairs within each ROI, or B) selecting the best-predicting layer for each voxel then averaging those scores within the ROI. If Method A is used, prefrontal ROI scores could potentially be systematically underestimated because most layers (early to intermediate layers) that perform poorly in behavioral linear readout would be anticipatedly to also poorly predict prefrontal activity (one would expect only the latest layers are relevant).
- Novelty?
The finding that larger models achieve higher brain alignment (Small < Base < Large < Giant; Figure 5) substantially replicates Huh et al. (2024, "Platonic Representation Hypothesis"), which tested multiple model families with varying sizes and observed similar scaling trends (with few exceptions where larger ≠ better). The present study's scope is even narrower (single architecture family DINOv3, only 4-5 size variants) to claim general principles about model size effects beyond DINOv3-specific properties. Though the region-specific analysis (Figure 5C showing disproportionate benefits for BA45 vs. V1) adds nuance

Minor prose issues:
Figure 3 axis- showing the relative score compared to the optimal score rather than raw score? consider clarifying in caption or axis label.
Line 212 missing line break (fMRI description runs into ROIs description)
Line 239 missing space after period
Line 301 extra space before “?”

**Questions:**

The temporal score emerges very early in training (~0.7% half-time), but this might be trivially expected from architecture alone rather than learned through training. The DINOv3 architecture inherently processes information from early-to-deep layers, which naturally corresponds to the brain's early-to-late temporal dynamics.
What is the temporal score for completely untrained (random) models? If it's already high, then the "emergence" during training is misleading.

The only difference training might make (given that the untrained ver seemed to give negative correlations)  is reducing the randomness in the layer-to-time mapping (i.e. randomly initialized model might just get a random assignment to each of the random-looking layers, thus resulting to low depth - time correlation score), not necessarily creating the “temporal alignment” de novo.
They mentioned in DI “deep layers of untrained models best predict early brain responses, which is backwards”, is this consistently observed or just suspicions (i.e. just inferred from ~0 negative correlation score)
Figure 6C shows that non-human-centric images (satellite, cellular) produce minimal prefrontal alignment.
Do you observe any alignment difference to prefrontal cortex (specific ROIs or relative alignment strength change) when the neural data were recorded under different behavioral tasks (e.g. those that involve judgement (recognition) (the fmri dataset) vs. passive viewing (the MEG dataset) ?
It would be interesting to see quantitative support for statements like "experience plays a larger role" in the Discussion.
Since some findings/ disentangling attempts align with previous study (and people’s intuition) - It would be interesting to dive into how these factors actually interact with each other. e.g.  Does the training amount effect depend on the model size? Is there a minimum model complexity (albeit difficult to define) where adding more training stops helping as much? Can you compensate for a smaller model by training it much longer, and if so, what's the tradeoff? if not, what were the minimal components of models necessary for training to be contributive to brain-alignment (representation level) ?
Will these findings always model/ training diet-specific? (will quantitative attempts always fail when we change these factors?) Can any general qualitative conclusion be drawn from this?

---

> ### Author Response · Authors · 2025-11-24
> **Generalization to other vision models: We now add 10 additional sets of results (among which a benchmark of 7 new models) and modify the manuscript**
>
> We thank Reviewer bhwe for their thorough review.
> We now add 10 additional sets of results  (among which a benchmark of 7 new models) and amend sections of the manuscript to address the issues raised and improve the paper. We uploaded the updated version of the paper, with modified text and the new figures.
>
> **1. Discussion on Dinov3 vs generalization to other vision models**
>
> This is indeed a very good point, the study was conducted specifically on DINOv3. To reflect this comment, we reproduced the results and 7 other diversified vision models and modified the manuscript.
>
> **1A. Clarification of the contribution**
>
>  First, we modified the title, intro, abstract and discussion of the paper to highlight the focus on DINOv3, as follows:
> - Title: To reflect the focus on DINOv3 and the work along DINOv3’s training, we modify the title to
> “Disentangling the Factors of Convergence between Brains and DINOv3”
> - Abstract:
> “We trained a family of self-supervised DINOv3 vision transformers”
> - Introduction:
> “we systematically train a variety of DINOv3 models”
> - Discussion:
> “The model–brain similarity increases consistently with larger DINOv3 architectures”
>
> **1B. Reproduction of key results on seven vision models - With trained models and untrained controls**
>
> Additionally, we reproduced the results of the manuscript across seven architectures and training objectives – including CNNs (ResNet-50, ConvNeXt-Large), a self-supervised ViT with different objective than DINOv3 (ViT-MAE, masked image reconstruction), a supervised ViT (ViT-L/16), and vision – language contrastive transformers (CLIP, SigLIP2, OWL-ViT).
> Adding scores obtained with random-weight models is a good control. These analyses compare brain score, spatial score, temporal score in untrained as well as fully trained versions of all these seven models.
> Results show that once trained:
> - The vast majority of vision models consistently have qualitatively similar encoding, spatial and temporal scores.
> - The trained vision models consistently have higher encoding, spatial and temporal scores than their untrained counterparts. These scores also improve across the training of DINOv3 in Fig. 3 of the manuscript.
> - The trained models tend to have similar encoding, spatial and temporal scores between models.
> - The scores of untrained vision models tend to vary more between models. A hypothesis for this behavior is that untrained models do not share the same inductive biases and start from very different initial points – yet they all eventually converge to similar representations and computational pathways, measurable through  encoding, spatial and temporal scores.
>
> To report these additional analyses, we add five figures S1, S2, S3, S4 and S5 in Appendix, presenting a comparative analysis of trained vs untrained versions of the 7 models studied, as well as brainplots and curves along time of encoding scores and maximally encoding layers.
>
> We adapted the Results sections to these new results as follows:
>
> 3.3.1 DINOv3-Brain similarity
>
> “To test for generalization of Encoding, Spatial and Temporal scores we reproduce these results on a variety of seven vision models including CNNs, supervised and self-supervised ViTs as well as vision-language contrastive transformers. We find similar scores for all three metrics, across all these models.”
>
> 3.3.2 What factors lead DINOv3 to become brain-like?
>
> “Across seven vision models with diverse architectures – CNNs, supervised ViTs and vision-language contrastive transformers –, trained models consistently show higher  encoding, spatial, and temporal scores than untrained models. Additionally, trained models tend to show convergent encoding, spatial, and temporal scores, whereas scores from untrained models vary more widely – likely reflecting differences in their inductive biases. These additional analyses are reported in Figs. S1–S5, comparing trained and untrained versions of all models.”
>
> **1C. Reasons for focusing on DINOv3 for analyses along trainings**
>
> We chose to study exclusively DINOv3 for 3 reasons:
> - It was the only model for which we could access checkpoints across the whole training from scratch.
> - We needed a common architecture for which to vary only one single parameter for each training (size, data type, etc.) to be able to disentangle factors of convergence with representations of the brain.
> - DINOv3 was the only model that we had the possibility to train multiple times from scratch, using controlled variants of the model and training on controlled sets of data from different types (cellular, satellite, human-centric). Self-supervision is also well suited regarding training on different types of data.
>
> **2. Discussion on ambiguity in ROI-level encoding scores**
>
> This is a good remark : to be able to compare scores across ROIs, we only compare the encoding score of the last layer. We now highlight this in the main text and legend of the paper.

---

> ### Author Response · Authors · 2025-11-24
> **Novel contributions, Initial values and Emergence of temporal and spatial alignments, Factors of convergence**
>
> **3. Results consistent with prior work and novel contributions**
>
> Thank you for this remark, our results are indeed consistent with prior work. However, here are our novel contributions:
>
> - **Controlled training**: We disentangle co-varying factors such as training data type, dataset size, and architectural depth.
> - **New similarity measure for hierarchical alignment**: We measure, beyond representations, similarities in hierarchies of processing – i.e. temporal and spatial scores.
> - **Analysis along training checkpoints**: We study how these brain-like properties emerge over the course of training. By disentangling these factors and showing, for example, that spatial alignment emerges earlier than temporal alignment, we provide new evidence that clarifies and extends the mechanisms underlying model–brain convergence.
> - **Cortical markers**: We find that the emergence of brain-like properties during training partially mirrors the developmental maturation of the human cortex from birth through early adulthood.
>
> Indeed, results are complementary with prior work:
>
> Huth et al (2024) examine how models converge with one another, without evaluating whether this convergence relates to the human brain. In contrast, we assess the extent to which such model–model convergence also aligns with neural representations and hierarchical organization in the human brain. This work is complementary with prior work, and its overall conclusion is consistent with this conjecture.
>
> We now make this contribution explicit in the discussion:
>
> “This study systematically disentangles co-varying factors such as training data type, dataset size, and architectural depth. Results show that size, training duration, and data type each shape the emergence of brain-like representations but also brain-like hierarchies, measured through spatial and temporal scores. Representations, temporal and spatial hierarchies all emerge - though at different pace across training. Our analyses complement prior work examining only model–model convergence (Huth et al, 2024), demonstrating that this convergence also extends onto human neural representations. Finally, we find that the emergence of brain-like properties during training follows the developmental maturation of the human cortex from birth through early adulthood.”
>
> **4. Minor prose issues:**
>
> Thank you for these good catches, we now fixed them.
>
> **5. Origin of the emergence of temporal alignment**
>
> “The only difference training might make is reducing the randomness”. This is an interesting point. Therefore, we change the Discussion of the manuscript to precise:
>
> “The factors that lead the temporal score to emerge are not entirely clear. However, the temporal score rises before the encoding score ( Fig. 3), suggesting that the encoding score alone does not solely explain the temporal score.”
>
> **6. Discussion on interacting factors and training amount effect depending on the model size**
>
> Regarding whether training amount effect depends on the model size, Figure 5A, B of the manuscript indicates that longer training cannot indeed compensate for a smaller architecture regarding encoding, temporal or spatial score, as all models from different sizes converge to different values. We find that larger models trained longer on human-centric data have stronger encoding, temporal or spatial scores. Future work will indeed be required to further boil down the minimal components necessary to brain-alignment, we now clarify this point in the discussion.
>
> **7. Discussion on impact of behavioral task vs passive viewing on the Prefrontal cortex**
>
> This is an interesting point, we currently cannot test this hypothesis by comparing the MEG and fMRI datasets as we do not possess the MEG sources, but we would like to study in a controlled way the impact of behavioral task vs passive viewing in future research.
>
> We modified the discussion as follows :
>
> “future work should assess how different tasks modulate the alignment between DINOv3 and the prefrontal cortex.”
>
> **8. Discussion spatial and temporal scores being initially negative**
>
> To answer this remark, we tested whether the vertical offsets of our logarithmic fits were significantly negative for spatial (-0.4) and temporal (-0.9) scores. We performed 1,000 permutations by shuffling the x-values and comparing the true offset to the resulting null distribution. The temporal score showed a significantly negative offset (p = 0.05), whereas the spatial score did not (p > 0.05). We now add these significance results in the discussion.
>
> Additionally, to complete the paper, we now add in Supplementary Figure S7 the non-normalized encoding, temporal and spatial scores of DINOv3 along training.

---

### Official Review · Reviewer_eWiA · 2025-11-04

**Soundness:** 3
**Presentation:** 3
**Contribution:** 2
**Rating:** 2
**Confidence:** 4

**Summary:**

The paper analyzes how variants of DINOv3, that were trained from scratch while independently varying model size, training duration, and image type, align with human brain activity. Then using  fMRI and MEG, they quantify alignment via three complementary metrics: an overall encoding score (linear predictability), a spatial score (hierarchical correspondence across cortical regions), and a temporal score (layer-to-latency correspondence). They find that all three factors matter, with the strongest brain similarity achieved by larger DINOv3 models trained longer on human-centric images, and that spatial and temporal hierarchies in the model mirror those observed in cortex.

This alignment follows a developmental trajectory during training: early layers quickly align with early visual areas and fast MEG responses, whereas later layers only align with higher-order (including prefrontal) regions after substantially more training. The speed (half-time) at which regions become linearly decodable from DINOv3 correlates with cortical properties

**Strengths:**

I think the paper is clear and easy to follow. It’s written in a straightforward way and focuses on a well-defined question: how certain factors within DINOv3 (model size, training time, and data domain) affect its ability to linearly predict human brain activity from fMRI and MEG. The methodology is clean, and the factorial setup makes the analysis intuitive.

This is a very exciting topic and the authors do a good job in  connecting representation learning with neuroscience. Also , they chose sensible axes for exploration. The three proposed metrics (encoding, spatial, and temporal) are complementary and help make the results interpretable. I like that they include both spatial (fMRI) and temporal (MEG) comparisons, which makes the findings richer and more biologically grounded. The “developmental trajectory” result is an interesting angle and ties well with biological plausibility.

**Weaknesses:**

The title should make clear that the analysis is only done on DINOv3, not on vision models in general.

I think the rationale for picking DINOv3 as the model to analyze isn’t well justified. There’s no comparison to other strong baselines that are already known to align well with brain data (e.g., ResNet, CORnet-S, CLIP, SimCLR). Without that, it’s hard to know if the observed effects are special to DINOv3 or just generic to large self-supervised ViTs.

Another issue is that the paper defines its own metrics and only applies them to DINOv3,  so it risks being a bit circular. It would be stronger if the same metrics were applied across multiple models to show whether DINOv3 is actually distinctive.

A control including the metrics  for untrained or random-weight models would be great. I think  including these would help confirm that the correlations aren’t just due to low-level image statistics or architectural biases (for example, ViTs already encode spatial frequency patterns similar to early visual cortex even before training).

Finally, the overall impact is somewhat limited because the findings: larger models trained longer on naturalistic images better align with brain data, are consistent with earlier work (e.g., Yamins & DiCarlo 2016; Schrimpf et al. 2018; Cichy et al. 2016). The novelty mainly comes from the within-family comparison and the correlations with cortical properties, so it would help if the authors clarified this and avoided overselling the generality.

**Questions:**

Scope clarification:
Why was the analysis restricted to DINOv3? Do you expect the same patterns (on size, training, and data domain) to hold for other self-supervised vision transformers or CNN-based models?

Choice of DINOv3:
What motivated selecting DINOv3 as the reference model for brain alignment? Was it chosen because of prior evidence of brain-like representations, or mainly because of its training flexibility?

Baselines:
Have you considered including other models (e.g., ResNet, CLIP, MAE, or supervised ViTs) in the same evaluation framework to check whether the observed effects are specific to DINOv3 or more general?

Metric validation:
How sensitive are your conclusions to the three metrics (encoding, spatial, temporal)? For instance, would a representational similarity analysis (RSA) or decoding-based metric lead to the same developmental trajectory findings?

Untrained controls:
Could you include results for untrained or partially trained DINOv3 checkpoints? This would help isolate the contribution of training versus architectural priors in explaining the fMRI/MEG alignment.

Spatial hierarchy measure:
The spatial score is based on Euclidean distance from V1, which is a pretty coarse proxy for cortical hierarchy. Did you test other spatial gradients (e.g., the principal sensory-to-transmodal gradient or visual hierarchy maps) to verify that the effect holds?

Interpretation of “human-centric data” effect:
How do you distinguish whether the advantage of human-centric images comes from their semantic content (ecological relevance) versus low-level statistics or augmentations used during training?

---

> ### Author Response · Authors · 2025-11-24
> **Generalization to other vision models: We now add 10 additional sets of results  (among which a benchmark of 7 new models) and modify the manuscript**
>
> We thank Reviewer eWiA for their thorough review.
> We now add 10 additional sets of results  (among which a benchmark of 7 new models) and amend sections of the manuscript to address the issues raised and improve the paper. We uploaded the updated version of the paper, with modified text and the new figures.
>
> **1. Discussion on Dinov3 vs generalization to other vision models**
>
> This is indeed a very good point, the study was conducted specifically on DINOv3. To reflect this comment, we reproduced the results and 7 other diversified vision models and modified the manuscript.
>
> **1A. Clarification of the contribution**
>
>  First, we modified the title, intro, abstract and discussion of the paper to highlight the focus on DINOv3, as follows:
> - Title: To reflect the focus on DINOv3 and the work along DINOv3’s training, we modify the title to
> “Disentangling the Factors of Convergence between Brains and DINOv3”
> - Abstract:
> “We trained a family of self-supervised DINOv3 vision transformers”
> - Introduction:
> “we systematically train a variety of DINOv3 models”
> - Discussion:
> “The model–brain similarity increases consistently with larger DINOv3 architectures”
>
> **1B. Reproduction of key results on seven vision models - With trained models and untrained controls**
>
> Additionally, we reproduced the results of the manuscript across seven architectures and training objectives – including CNNs (ResNet-50, ConvNeXt-Large), a self-supervised ViT with different objective than DINOv3 (ViT-MAE, masked image reconstruction), a supervised ViT (ViT-L/16), and vision – language contrastive transformers (CLIP, SigLIP2, OWL-ViT).
> Adding scores obtained with random-weight models is a good control. These analyses compare brain score, spatial score, temporal score in untrained as well as fully trained versions of all these seven models.
> Results show that once trained:
> - The vast majority of vision models consistently have qualitatively similar encoding, spatial and temporal scores.
> - The trained vision models consistently have higher encoding, spatial and temporal scores than their untrained counterparts. These scores also improve across the training of DINOv3 in Fig. 3 of the manuscript.
> - The trained models tend to have similar encoding, spatial and temporal scores between models.
> - The scores of untrained vision models tend to vary more between models. A hypothesis for this behavior is that untrained models do not share the same inductive biases and start from very different initial points – yet they all eventually converge to similar representations and computational pathways, measurable through  encoding, spatial and temporal scores.
>
> To report these additional analyses, we add five figures S1, S2, S3, S4 and S5 in Appendix, presenting a comparative analysis of trained vs untrained versions of the 7 models studied, as well as brainplots and curves along time of encoding scores and maximally encoding layers.
>
> We adapted the Results sections to these new results as follows:
>
> 3.3.1 DINOv3-Brain similarity
>
> “To test for generalization of Encoding, Spatial and Temporal scores we reproduce these results on a variety of seven vision models including CNNs, supervised and self-supervised ViTs as well as vision-language contrastive transformers. We find similar scores for all three metrics, across all these models.”
>
> 3.3.2 What factors lead DINOv3 to become brain-like?
>
> “Across seven vision models with diverse architectures – CNNs, supervised ViTs and vision-language contrastive transformers –, trained models consistently show higher  encoding, spatial, and temporal scores than untrained models. Additionally, trained models tend to show convergent encoding, spatial, and temporal scores, whereas scores from untrained models vary more widely – likely reflecting differences in their inductive biases. These additional analyses are reported in Figs. S1–S5, comparing trained and untrained versions of all models.”
>
> **1C. Reasons for focusing on DINOv3 for analyses along trainings**
>
> We chose to study exclusively DINOv3 for 3 reasons:
>
> - It was the only model for which we could access checkpoints across the whole training from scratch.
> - We needed a common architecture for which to vary only one single parameter for each training (size, data type, etc.) to be able to disentangle factors of convergence with representations of the brain.
> - DINOv3 was the only model that we had the possibility to train multiple times from scratch, using controlled variants of the model and training on controlled sets of data from different types (cellular, satellite, human-centric). Self-supervision is also well suited regarding training on different types of data.

---

> ### Author Response · Authors · 2025-11-24
> **Novel contributions, metric validation, reproduction of the spatial score on unimodal-to-transmodal gradient, adding of a paragraph in discussion regarding human-centric data**
>
> **2. Results consistent with prior work**
>
> Thank you for this remark, our results are indeed consistent with prior work. However, here are our novel contributions:
> - **Controlled training**: We disentangle co-varying factors such as training data type, dataset size, and architectural depth.
> - **New similarity measure for hierarchical alignment**: We measure, beyond representations, similarities in hierarchies of processing – i.e. temporal and spatial scores.
> - **Analysis along training checkpoints**: We study how these brain-like properties emerge over the course of training. By disentangling these factors and showing, for example, that spatial alignment emerges earlier than temporal alignment, we provide new evidence that clarifies and extends the mechanisms underlying model–brain convergence.
> - **Cortical markers**: We find that the emergence of brain-like properties during training partially mirrors the developmental maturation of the human cortex from birth through early adulthood.
>
> We now highlight our contribution more clearly in the manuscript:
>
> “This study systematically disentangles co-varying factors such as training data type, dataset size, and architectural depth. Results show that size, training duration, and data type each shape the emergence of brain-like representations but also brain-like hierarchies, measured through spatial and temporal scores. Representations, temporal and spatial hierarchies all emerge - though at different pace across training. Our analyses complement prior work examining only model–model convergence (Huth et al, 2024), demonstrating that this convergence also extends onto human neural representations. Finally, we find that the emergence of brain-like properties during training follows the developmental maturation of the human cortex from birth through early adulthood.”
>
> **3. Discussion on metric validation (encoding, decoding, RSA)**
>
> Thank you for raising this important question. We chose encoding as base for our three metrics (encoding, spatial, temporal scores) because decoding metrics could not be directly compared across models in our setting. Indeed, each model defines a different representational space, making the decoded targets fundamentally mismatched. RSA would indeed work in theory but would be too costly computationally for our large datasets (e.g., 70k × 70k RDMs).
>
> We now amend the Methods in the manuscript to express why we rely on encoding-, spatial-, and temporal-alignment metrics, which provide a consistent basis for comparing developmental trajectories across architectures and data types:
>
> “We rely on encoding as the basis for our three metrics (encoding, spatial, and temporal scores), rather than decoding, as decoding metrics cannot be meaningfully compared across models with different architectures and representational spaces. Following the rationale of (Naselaris et al, 2011), encoding provides an interpretable mapping from model features to neural responses, comparable across architectures and training regimes.”
>
> **4. Reproduction of the spatial score based on the sensory-to-transmodal gradient map**
>
> Reproducing the spatial score through a finer gradient was indeed a very good point. We partially reproduced the spatial scores results obtained on DINOv3 using the sensory-to-transmodal gradient map from Margulies et al. (2016, PNAS) for the first and last checkpoints of DINOv3 across training (0.1% and 100% of training) and we now report the results of this alternative spatial score in Supplementary figure S6 in the manuscript. This alternative spatial score increases across training from -0.32 to 0.13 (1st to last checkpoint), similarly to the regular spatial scores which increases from -0.4 to 0.4 across training (1st to last checkpoint).
>
> We report these results in the Result section of the paper as follows:
>
> 3.3.2 What factors lead DINOv3 to become brain-like? Across training
>
> “We partially reproduce the evolution of spatial score using the value of each ROI along the sensory-to-transmodal gradient map from Margulies et al. (2016) instead of the euclidean distance of this ROI from V1, see Supplementary Figure S6. We obtain a similar increase across training from -0.32 to 0.13 (1st to last checkpoint).”
>
> **5. Discussion on the “human-centric data” effect coming from semantic content versus low-level statistics**
>
> This is a good remark, we modify the discussion to extend on this point:
>
> “Our findings indicate that models trained on human-centric images tend to develop representations that more closely resemble those of the human visual system. However, it remains unclear whether this advantage reflects low-level image statistics (e.g., natural color and texture distributions) or higher-level semantic properties typical of human experience. Distinguishing between these factors will require future research, for example by evaluating brain responses from participants watching controlled non-human centric images.”

---

> > ### Comment · Reviewer_eWiA · 2025-11-24
> >
> > I want to thank the authors for their  thorough and thoughtful rebuttal. I appreciate the substantial additionsmade, especially the inclusion of seven new baseline models, untrained controls, and the alternative spatial hierarchy analysis using the Margulies gradient. These revisions significantly strengthen the manuscript and address most of the concerns I raised. I also find the clarified framing around the DINOv3 focus (title, abstract, introduction) appropriate and helpful.
> >
> > The added comparative analyses (trained vs. untrained) and the convergence of spatial/temporal scores across diverse architectures make the generalization claims much clearer. The explanation for relying on encoding-based metrics also reads more convincingly after your methodological clarifications.
> >
> > One remaining question for the discussion:
> > Given that many of the newly added baselines achieve similar encoding, spatial, and temporal scores once trained, how should readers interpret the specific contribution of DINOv3’s training trajectory analyses (size, data type, human-centric advantage, etc.)? In other words, do you view these developmental findings as properties of DINOv3 specifically, or do you expect similar developmental trajectories (early low-level → late high-level alignment) to hold for other models if full training checkpoints were available?
> >
> > Understanding this distinction would help position the broader significance of the training-trajectory portion of the paper.
> >
> > Thank you again for the comprehensive response and the added results.

---

> ### Author Response · Authors · 2025-11-25
>
> We thank Reviewer eWiA for their thorough and thoughtful comments. Indeed, given the convergence of encoding, spatial, and temporal scores for the eight studied models once trained, we can hypothesize that the results obtained through DINOv3’s checkpoints – size, data type, and training duration steering convergence toward brain-like properties across the three metrics and developmental trajectories (early low-level → late high-level alignment) – could be generalizable across more models.
>
> We now add this interesting remark in the Discussion:
>
> “Results show that size, training duration, and data type each shape the emergence of brain-like representations but also brain-like hierarchies, measured through spatial and temporal scores. Representations, temporal and spatial hierarchies all emerge – though at different pace across training. When comparing untrained and trained versions of seven diversified vision models, we find that the vast majority of them consistently develop qualitatively similar encoding, spatial and temporal scores as DINOv3.[…] Finally, we find that the emergence of brain-like properties during training follows the developmental maturation of the human cortex from birth through early adulthood. **Although these developmental trajectories have not been tested in other models, the convergence of encoding, spatial, and temporal scores for eight diversified vision models may suggest that these additional results regarding DINOv3 are generalizable to other models. Future research will allow to test this hypothesis.**”

---

### Meta-Review · Area_Chair_nVjx · 2026-01-06

**Summary:**

Two of the reviewers gave positive scores (6, 8) while two of the reviewers gave negative scores (2, 4). The main concerns are that the analyses are only for a single model (DINOv3) with a new performance metric that may be tailed for this model and that similar results have already been shown.

**Reviewer Concerns:**

The authors repeated their analyses on 7 additional vision models, which addresses the concern that the original analyses were only performed on a single model. The authors also clarified that their analyses disentangle what factors (dataset properties, model size) contribute to brain-model alignment, which differentiates their submission from previous works. I believe the primary concerns that were raised by the reviewers were addressed.

**Reviewer Scores:**

I believe the reviewers who gave positive scores would have maintained their scores. I believe that Reviewer eWiA would have raised their score from a 2 to 6 and I believe that Reviewer bhwe would have raised their score from a 4 to 6.

---

### Decision · Program_Chairs · 2026-01-26

Accept (Poster)